# HAMLET: SWITCH YOUR VISION-LANGUAGE-ACTION MODEL INTO A HISTORY-AWARE POLICY

**Myungkyu Koo**[*1]  **Daewon Choi**[*1]  **Taeyoung Kim**[1]  **Kyungmin Lee**[1]
**Changyeon Kim**[1]  **Younggyo Seo**[†2]  **Jinwoo Shin**[†1,3]
[1]KAIST  [2]UC Berkeley  [3]RLWRLD
{jameskoo0503, daeone0920}@kaist.ac.kr

## ABSTRACT

Inherently, robotic manipulation tasks are history-dependent: leveraging past context could be beneficial. However, most existing Vision-Language-Action models (VLAs) have been designed without considering this aspect, *i.e.*, they rely solely on the current observation, ignoring preceding context. In this paper, we propose HAMLET, a scalable framework to adapt VLAs to attend to the historical context during action prediction. Specifically, we introduce *moment tokens* that compactly encode perceptual information at each timestep. Their representations are initialized with time-contrastive learning, allowing them to better capture temporally distinctive aspects. Next, we employ a lightweight *memory module* that integrates the moment tokens across past timesteps into memory features, which are then leveraged for action prediction. Through empirical evaluation, we show that HAMLET successfully transforms a state-of-the-art VLA into a history-aware policy, especially demonstrating significant improvements on long-horizon tasks that require historical context. In particular, on top of GR00T N1.5, HAMLET achieves an average success rate of 76.4% on history-dependent real-world tasks, surpassing the baseline performance by 47.2%. Furthermore, HAMLET pushes prior art performance from 64.1% to 66.4% on RoboCasa Kitchen (100-demo setup) and from 95.6% to 97.6% on LIBERO, highlighting its effectiveness even under generic robot-manipulation benchmarks. Project page: https://myungkyukoo.github.io/hamlet/

## 1 INTRODUCTION

Vision-Language-Action models (VLAs; Zitkovich et al. 2023; Kim et al. 2024; Black et al. 2025; Pertsch et al. 2025; Li et al. 2024a; Qu et al. 2025; Bjorck et al. 2025b) have shown their promise in robotic policy learning by leveraging large-scale pre-trained Vision-Language Models (VLMs; Beyer et al. 2024; Chen et al. 2023; Driess et al. 2023; Karamcheti et al. 2024; Touvron et al. 2023) with diverse robot-specific datasets (Walke et al., 2023; O'Neill et al., 2024; Khazatsky et al., 2024). They typically adopt a single-frame assumption, predicting each action solely from the current observation. However, such reliance on a current *snapshot* fundamentally limits their capability, since robotic manipulation tasks are intrinsically history-dependent. For instance, consider a simple scenario of placing an object on a table. The decision to move the arm depends on prior context—specifically, whether the object has already been grasped. When restricted to the current frame, the policy may struggle to determine the proper next action, particularly if the object is occluded.

Despite being a desirable property, incorporating history-awareness during pre-training is viewed as a costly design choice. A major challenge is that leveraging historical context incurs substantial computational overhead. For example, we observe that naïvely appending only four additional past observation frames to the VLA input slows down the forward pass by ∼35% and increases peak memory consumption by ∼3.6× (see *Multi-frame* in Table 4). In particular, the inflated memory footprint further restricts scalability by reducing feasible batch sizes compared to the single-frame setting. Together, these observations raise a key research question: *How can we integrate history-awareness into pre-trained VLAs without resorting to costly pre-training from scratch?*

---

[*]Equal contribution.
[†]Equal advising.

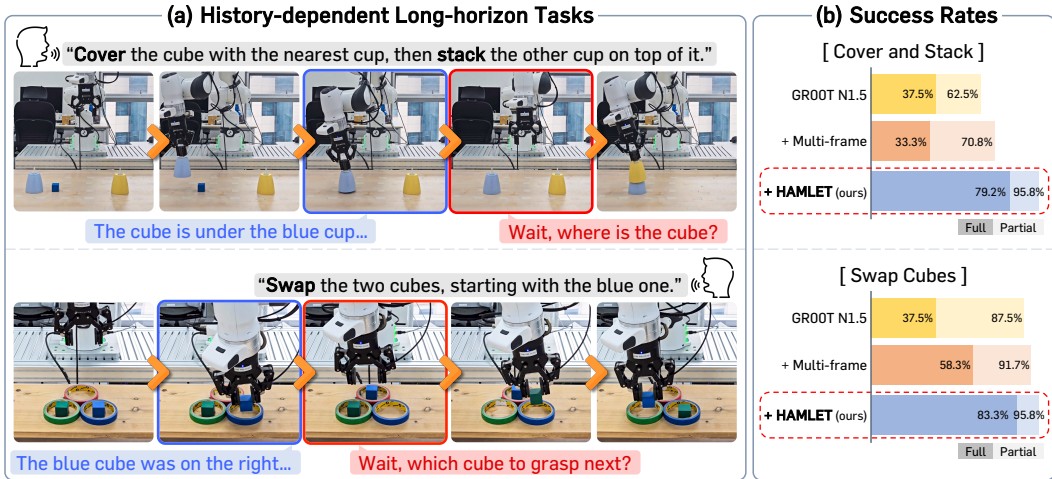

Figure 1: **Examples of history-dependent long-horizon tasks.** (a) Real-world tasks that involve cases such as object occlusion (upper) or multi-step reasoning (lower) are inherently *non-Markovian*: proper actions cannot be determined from the current observation alone. (b) Success rates on these tasks show that HAMLET significantly outperforms both GR00T N1.5 and the naïve multi-frame baseline, demonstrating its ability to leverage historical context for reliable long-horizon control.

To tackle this problem, we propose **HAMLET**, a fine-tuning framework for VLAs that introduces **H**istory-**A**ware **M**emory with **LE**arned **T**okens. Our framework consists of two components: (a) *moment tokens*, which summarize the instantaneous VLM representations at each timestep, and (b) a *memory module*, which consolidates moment tokens across different timesteps to produce a temporally informed condition for action prediction. The moment tokens are appended to the VLM input at each timestep and initialized with time-contrastive learning (Sermanet et al., 2018), which encourages distinctiveness across timesteps. This initialization enables the moment tokens to emphasize task-relevant dynamics while suppressing redundant information such as static backgrounds (see Figure 4 for details). Building on this, we incorporate a lightweight memory module that stores and integrates moment token representations across timesteps. This design is motivated by the observation that not all moments are equally informative; treating every timestep with equal importance can introduce redundancy and obscure critical cues (see *Moment Concat.* in Table 5c).

To validate the effectiveness and generality of HAMLET, we conduct comprehensive experiments across both real-world and simulation environments. We first evaluate HAMLET on the long-horizon, real-world tasks that require reasoning over past trajectories. We show that HAMLET improves performance by 47.2% over the naïvely fine-tuned VLA, which demonstrates the effectiveness of exploiting historical information for real-world robot policy learning. We further examine the generality and applicability of HAMLET across different VLA backbones. When fine-tuning GR00T N1.5 (Bjorck et al., 2025a) on the RoboCasa (Nasiriany et al., 2024) Kitchen dataset, HAMLET achieves an average success rate of 66.4%, compared to 64.1% for the baseline. Similarly, when applied to CogACT (Li et al., 2024a) on the SimplerEnv-Bridge (Li et al., 2024c) dataset, HAMLET attains 63.5%, substantially improving over the baseline performance of 52.1%. These results highlight that incorporating history-awareness consistently yields benefits across diverse VLA policies, and that HAMLET provides consistent improvements in a flexible, plug-in manner.

**Contributions.** Our contributions are as follows:

- Motivated by VLAs' reliance on the current observation alone, we propose HAMLET, an easily integrable framework that enables history-awareness in pre-trained VLAs.

- We introduce *moment tokens*, initialized with time-contrastive learning, to capture key temporal cues at each timestep. Building on this, we design a lightweight *memory module* that selectively aggregates these tokens across timesteps to produce history-aware features for action prediction.

- We validate HAMLET across both real-world and simulation benchmarks, achieving substantial gains over state-of-the-art baselines. By alleviating backbone models' reliance on the current observation, HAMLET delivers consistent improvements, especially with the strongest benefits

on long-horizon tasks. Importantly, its backbone-agnostic design allows seamless and efficient integration into diverse VLAs without requiring any additional large-scale pre-training.

## 2 RELATED WORKS

**Vision-Language-Action models (VLAs).** Since collecting high-quality and large-scale robot datasets is challenging, traditional robot learning (Shafiullah et al., 2022; Cui et al., 2022; Chi et al., 2023; Lee et al., 2024) has relied on task-specific data, which cover only a narrow distribution of environments, objects, and task instructions. To address this limitation, recent studies (Zitkovich et al., 2023; Driess et al., 2023; Kim et al., 2024; Pertsch et al., 2025; Bjorck et al., 2025a; Li et al., 2024a; Black et al., 2025) propose building generalist robot policies by utilizing internet-scale VLM priors to low-level action prediction, thereby transferring semantic understanding to robotic control. Early VLAs (Zitkovich et al., 2023; Kim et al., 2024; Driess et al., 2023) discretize the continuous action space and directly predict actions as tokens, demonstrating that pre-trained VLMs can be adapted to robot control. More recent approaches (Black et al., 2025; Bjorck et al., 2025b; Li et al., 2024a) leverage VLM representations to condition action experts on diffusion or flow-matching, enabling more accurate action prediction. For example, Black et al. (2025) conditions a flow-matching head on VLM features to produce action chunks per step, and Li et al. (2024a) systematically compares action modules, finding that diffusion action transformers scale favorably when conditioned on VLM representations. However, these models typically generate actions based only on the current observation, limiting their capability to accurately recognize the current state and determine precise actions. In this work, we focus on recent pre-trained VLAs with diffusion-based action heads, and demonstrate how our framework can enhance their performance by incorporating historical context.

**Memory architectures.** Traditionally, long-horizon tasks have been framed as non-Markovian problems, requiring policies to integrate memory to leverage past observations and actions. In reinforcement learning, recurrent policies (Hausknecht & Stone, 2015) and later Transformer-based variants (Parisotto et al., 2020) introduced memory mechanisms that improved performance on partially observable and long-horizon benchmarks. In natural language processing, explicit memory architectures have been widely explored, ranging from end-to-end memory networks (Sukhbaatar et al., 2015) to retrieval-based approaches (Khandelwal et al., 2019; Lewis et al., 2020), all designed to enhance long-context reasoning and knowledge integration. These advances highlight the importance of dedicated memory modules for tasks requiring extended temporal or contextual awareness. In robotics, prior works (Wen et al., 2020; Seo et al., 2023; Li et al., 2024b; Huang et al., 2025) have incorporated past observations into policy learning, but most approaches (Wen et al., 2020; Seo et al., 2023) rely on simple policy architectures (*e.g.*, MLPs or RNNs) with limited generalizability and require training from scratch (Li et al., 2024b; Huang et al., 2025), leading to heavy computational cost. Consequently, memory design for large-scale pre-trained VLAs remains under-explored. As a concurrent effort, Shi et al. (2025) proposes architectures inspired by human memory systems, trained from scratch, and demonstrates promising improvements on temporally dependent tasks. Distinct from this line of work, our approach augments pre-trained VLAs with a few learnable tokens and a lightweight memory module, allowing them to attend to history-awareness without retraining.

## 3 METHOD

In this section, we introduce HAMLET, a plug-in framework that adapts pre-trained Vision-Language-Action models (VLAs) to attend the historical context. Formally, let $\mathbf{o}_t = [\mathbf{I}_t^1, \ldots, \mathbf{I}_t^n]$ be the sequence of visual observations at timestep $t$, and $\mathbf{c}$ be a task instruction. The VLA $\mathcal{F}_\theta$ processes these inputs through its Vision-Language Model (VLM) backbone to obtain a hidden representation $\mathbf{h}_t$:

$$\mathbf{h}_t = \mathcal{F}_\theta(\mathbf{o}_t, \mathbf{c}). \tag{1}$$

Then, the representation $\mathbf{h}_t$ is used as a condition for the action expert $\mathcal{A}_\psi$ to predict a sequence of $k$ future actions, namely *action chunking* (Zhao et al., 2023; Chi et al., 2023):

$$[\mathbf{a}_t, \mathbf{a}_{t+1}, \ldots, \mathbf{a}_{t+k-1}] = \mathcal{A}_\psi(\mathbf{h}_t, \mathbf{s}_t), \tag{2}$$

where $\mathbf{s}_t$ denotes the robot's proprioceptive state at timestep $t$. After executing the predicted action sequence, the environment returns a new observation $\mathbf{o}_{t+k}$, which serves as the next input to the VLA. Here, our goal is to augment $\mathbf{h}_t$ with informative representations from previous timesteps

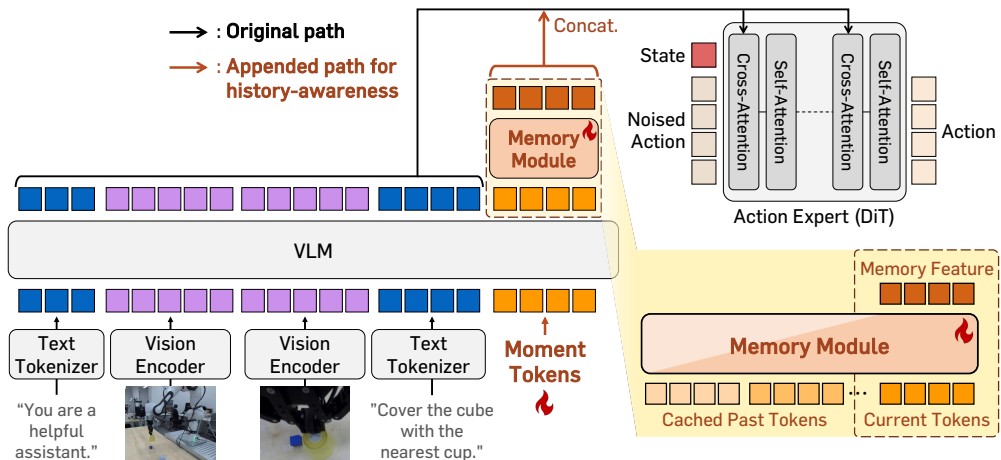

Figure 2: **An overview of HAMLET.** Building on a pre-trained VLA, HAMLET adds two key components: *moment tokens*, appended to the VLM input and initialized with time-contrastive learning to capture task-relevant representations at each timestep, and a lightweight *memory module* that aggregates these tokens across timesteps for history-aware action prediction.

(*i.e.*, $\mathbf{o}_{t-k}, \mathbf{o}_{t-2k}, \ldots$), to enable the action expert to effectively utilize long-horizon context. To achieve this goal, we propose two complementary components: (i) *moment tokens*, which compress the information at each timestep into a compact representation (see Section 3.1), and (ii) a *memory module*, which aggregates moment tokens across timesteps to yield a temporally-enriched condition for action prediction (see Section 3.2). The overall framework is illustrated in Figure 2.

## 3.1 CONTEXT COMPRESSION VIA MOMENT TOKENS

To leverage historical context, we first consider how to effectively store information from each timestep. Retaining the raw visual observation $\mathbf{o}_t$ at every timestep is suboptimal: it incurs high latency and memory costs (see Table 4 in Section 4.3) and often contains redundant or static elements that might provide irrelevant signal (Xu et al., 2025; Yang et al., 2025). To address this, we propose to compress the observation at each timestep into a concise representation that preserves task-relevant information while filtering out redundancy. This motivates the our design of *moment tokens*.

**Use of moment tokens.** At each timestep $t$, we append a set of learnable *moment tokens* $\mathbf{m}_t \in \mathbb{R}^{n_{\mathrm{m}} \times d}$ to the input sequence of the VLM, where $n_{\mathrm{m}}$ is the number of tokens and $d$ is the embedding dimension. Given a visual observation $\mathbf{o}_t$ and task instruction $\mathbf{c}$, we append moment token $\mathbf{m}_t$ to them and feed the combined input to the VLM encoder $\mathcal{F}_\theta$ to produce a hidden representation:

$$[\mathbf{h}_t; \mathbf{m}'_t] = \mathcal{F}_\theta([\mathbf{o}_t, \mathbf{c}; \mathbf{m}_t]), \tag{3}$$

where $[\,\cdot\,;\,\cdot\,]$ denotes concatenation of token sequences. Along with the hidden states $\mathbf{h}_t$, we extract the representation $\mathbf{m}'_t$ of moment tokens, which act as compact, context-aware summaries of the scene at timestep $t$. Due to the causal attention operator in the VLM, moment tokens attend to the current visual observation $\mathbf{o}_t$ and the task instruction $\mathbf{c}$. As a result, $\mathbf{m}'_t$ serves as a compressed representation of each timestep, which subsequently be stored and aggregated by the memory module.

**Time-contrastive learning.** To encourage moment tokens to encode temporally discriminative cues at each timestep, we draw inspiration from time-contrastive network (Sermanet et al., 2018; Nair et al., 2022; Ma et al., 2023), while adapting the design of positive pairs to image-augmented samples with photometric, blur, noise, and occlusion perturbations. For a trajectory $[\mathbf{o}_0, \ldots, \mathbf{o}_{T-1}]$ and task instruction $\mathbf{c}$, we extract moment token representations $\mathbf{m}'_t$ at each timestep using Eq. (3).

To construct the contrastive objective, for each timestep $t$ we form an anchor from the current observation $\mathbf{o}_t$. We then generate a positive $\mathbf{z}_t^+$ from an augmented view of the same observation and a hard negative $\mathbf{z}_t^-$ from a different timestep $t' \neq t$ within the same trajectory. Formally, let $\mathbf{z}_t = g(\mathbf{m}'_t)$ denote the projected moment-token representation produced by a projection head $g(\cdot)$.

We then optimize the following time-contrastive learning objective:

$$\mathcal{L}_{\text{TCL}}(\mathbf{z}_t, \mathbf{z}_t^+) = -\sum_{t=1}^{B} \log \frac{\exp\big(\text{sim}(\mathbf{z}_t, \mathbf{z}_t^+)/\tau\big)}{\exp\big(\text{sim}(\mathbf{z}_t, \mathbf{z}_t^+)/\tau\big) + \exp\big(\text{sim}(\mathbf{z}_t, \mathbf{z}_t^-)/\tau\big)}, \tag{4}$$

where $\text{sim}(\mathbf{a}, \mathbf{b})$ denotes cosine similarity and $\tau$ is a temperature hyperparameter. The summation indexes the $B$ anchors in the minibatch. This initialization encourages the moment tokens to align with representations from the same timestep while remaining discriminative across timesteps, enabling $\mathbf{m}_t$ to capture unique, timestep-specific cues while suppressing static components. During this stage, we freeze the VLM $\mathcal{F}_\theta$ to ensure that the loss does not distort its pre-trained representation.

## 3.2 MEMORY CONSOLIDATION VIA MEMORY MODULE

We now present a memory module to integrate the moment token representations $\mathbf{m}_t'$ across timesteps for action prediction. We observe that simply concatenating these representations does not directly improve performance (see *Moment Concat.* in Table 5c). Hence, we employ a lightweight Transformer $\mathcal{M}_\phi$ that selectively attends to informative past timesteps while ignoring less relevant ones.

**Design of memory module.** To incorporate historical context beyond current timestep, we introduce a memory module $\mathcal{M}_\phi$ that aggregates moment token representations across timesteps. Specifically, we employ a shallow Transformer (Vaswani et al., 2017) that attends over past moment tokens via causal self-attention. We form a history matrix by stacking the most recent $T$ moment tokens:

$$\mathbf{M}' = [\mathbf{m}_{t-k(T-1)}'; \ \dots; \ \mathbf{m}_{t-k}'; \ \mathbf{m}_t'] \in \mathbb{R}^{L \times d}, \tag{5}$$

where $k$ is the action-chunk length from Eq. (2), $T$ is the history length, and $L = T \cdot n_{\text{m}}$ is the total number of tokens. From $\mathbf{M}'$, the memory module applies standard self-attention:

$$\mathbf{Q} = \mathbf{M}'\mathbf{W}_q, \quad \mathbf{K} = \mathbf{M}'\mathbf{W}_k, \quad \mathbf{V} = \mathbf{M}'\mathbf{W}_v, \quad \mathbf{H} = \text{softmax}\Big(\frac{\mathbf{Q}\mathbf{K}^\top}{\sqrt{d}} + \mathbf{C}\Big)\mathbf{V}, \tag{6}$$

where $\mathbf{C}$ is a causal mask ensuring the proper encoding for sequential trajectory. $\mathbf{H}$ is mapped through the Transformer's output projection, producing $\tilde{\mathbf{M}}' \in \mathbb{R}^{L \times d}$. Then, we take the last $n_{\text{m}}$ rows of $\tilde{\mathbf{M}}'$, denoted $\tilde{\mathbf{m}}_t'$, as the *history-augmented* moment token representation for timestep $t$.

**Integration into action prediction.** The history-augmented feature $\tilde{\mathbf{m}}'$ is concatenated with the original VLM representation $\mathbf{h}_t$ and fed into the action expert $\mathcal{A}_\psi$ to predict the next $k$ actions.

$$[\mathbf{a}_t, \mathbf{a}_{t+1}, \dots, \mathbf{a}_{t+k-1}] = \mathcal{A}_\psi([\mathbf{h}_t; \tilde{\mathbf{m}}'], \mathbf{s}_t). \tag{7}$$

The overall training procedure follows that of standard VLA models, where the pipeline is trained end-to-end with the action prediction loss (Bjorck et al., 2025b; Li et al., 2024a; Black et al., 2025).

## 4 EXPERIMENTS

We design our experiments to investigate the following questions:

- Does applying HAMLET to existing VLAs enhance performance on long-horizon, real-world tasks that require reasoning over past trajectories? (Table 1 in Section 4.2)
- Is HAMLET also beneficial on generic robot-manipulation benchmarks? (Table 2, 3 in Section 4.2)
- Can HAMLET be seamlessly applied across different pre-trained VLAs? (Table 3 in Section 4.2)
- How does HAMLET perform in terms of computational overhead, effective design choices, and transferability to unseen datasets? (Table 4, 5, 6 in Section 4.3, respectively)

## 4.1 EXPERIMENTAL SETUPS

**Datasets.** We evaluate HAMLET on real-world tasks that require reasoning over past trajectories, as well as on diverse simulation benchmarks (Figure 5). In the real-world environment, we design three handcrafted tabletop tasks: (i) *Pick-and-Place Twice*, where the robot moves a cube between two sides twice; (ii) *Cover-and-Stack*, where the robot covers a cube with one cup and then stacks it with another; and (iii) *Swap Cubes*, where the robot swaps the positions of two cubes using an auxiliary

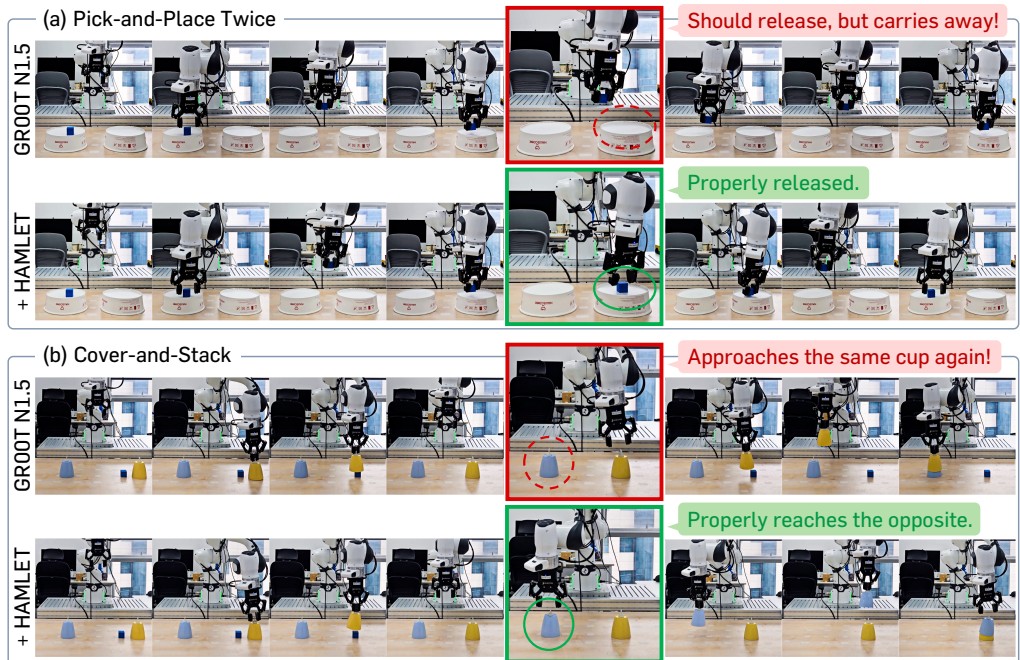

Figure 3: **Example rollouts of real-world tasks.** We present example rollouts executed by our HAMLET and GR00T N1.5, respectively. While HAMLET predicts proper next actions, in (a) GR00T N1.5 is confused about whether it should lift or release the cube, and in (b) it fails to identify which cup has a cube underneath, due to the absence of historical context.

Table 1: **Real-world evaluation results.** We report the success rate (%, over 24 trials per task) on three real-world tasks: partial success rates for columns (*PnP Once, Cover Cube, Stage Cube*), and 'Success' for full completion. **Bold** and underline indicate the best and runner-up results, respectively.

| Method | History? | Pick-and-Place Twice | | Cover-and-Stack | | Swap Cubes | | |
| --- | --- | --- | --- | --- | --- | --- | --- | --- |
| | | *PnP Once* | Success | *Cover Cube* | Success | *Stage Cube* | Success | Avg. |
| $\pi_0$ | ✗ | 54.2 | 25.0 | 87.5 | 58.3 | 83.3 | 12.5 | 31.9 |
| $\pi_0$-FAST | ✗ | 37.5 | 20.8 | 54.2 | 12.5 | 66.7 | 4.2 | 12.5 |
| GR00T N1 | ✗ | 54.2 | 25.0 | 79.2 | 33.3 | 75.0 | 33.3 | 30.6 |
| GR00T N1.5 | ✗ | 54.2 | 12.5 | 62.5 | 37.5 | 87.5 | 37.5 | 29.2 |
| + Multi-frame | ✓ | 79.2 | 45.8 | 70.8 | 33.3 | 91.7 | 58.3 | 45.8 |
| **+ HAMLET (Ours)** | ✓ | **91.7** | **66.7** | **95.8** | **79.2** | **95.8** | **83.3** | **76.4** |

site. For the simulation environment, we conduct experiments on three widely-used benchmarks: RoboCasa (Nasiriany et al., 2024) Kitchen, LIBERO (Liu et al., 2023) and SimplerEnv-Bridge (Li et al., 2024c), which consist of multi-step household manipulation tasks spanning diverse objects and configurations. Further details including real-world robot setups are provided in Section A.2.

**Baselines.** We design baseline comparisons according to the target benchmark. We primarily evaluate on GR00T N1.5 (Bjorck et al., 2025b) and further assess generalization on CogACT (Li et al., 2024a). For real-world tasks and simulation benchmarks (RoboCasa Kitchen and LIBERO), we compare HAMLET on GR00T N1.5 with representative baselines: $\pi_0$ (Black et al., 2025), $\pi_0$-FAST (Pertsch et al., 2025), and GR00T N1 (Bjorck et al., 2025b). On SimplerEnv-Bridge, we evaluate HAMLET on CogACT against the reported performances of OpenVLA (Kim et al., 2024), Octo (Team et al., 2024), RoboVLM (Liu et al., 2025) and SpatialVLA (Qu et al., 2025). For comparison with methods that utilize historical context, we implement the multi-frame baseline, which stores past observation frames and concatenates them into the VLA input (see Section A.3 for details).

**Implementation details.** We apply HAMLET to each VLA following the original fine-tuning setup of that model, without heuristic hyperparameter tuning. Instead, we adopt the training configurations specified for each backbone (*e.g.*, learning rate, optimizer, and freezing of the VLM backbone). By

Table 2: **Simulation benchmark results on GR00T N1.5.** We compare HAMLET with baseline methods on RoboCasa Kitchen and LIBERO. For RoboCasa Kitchen, we report the average success rate (%) across 24 tasks with models trained using 30, 100, or 300 demonstrations per task. For LIBERO, each metric is the average success rate (%) across 10 tasks per suite, with training performed jointly on all suites. All the results are reproduced by us, except for those of GR00T N1 on RoboCasa Kitchen. **Bold** and underline indicate best and runner-up results, respectively.

| Method | RoboCasa Kitchen (# of demos) | | | LIBERO (task suite) | | | | |
| | 30 | 100 | 300 | Spatial | Object | Goal | Long | Avg. |
|---|---|---|---|---|---|---|---|---|
| $\pi_0$ | 47.8 | 58.7 | 62.5 | 97.2 | 97.2 | 93.6 | 89.2 | 94.3 |
| $\pi_0$-FAST | 29.8 | 60.2 | 63.6 | 96.0 | 96.4 | 91.6 | 85.0 | 92.3 |
| GR00T N1 | 17.4 | 32.1 | 49.6 | 95.6 | 97.6 | 94.2 | 89.6 | 94.3 |
| GR00T N1.5 | 47.8 | 62.6 | 64.1 | 98.2 | 99.4 | 97.2 | 87.8 | 95.6 |
| + Multi-frame | 44.0 | 59.3 | 60.8 | 81.4 | 97.2 | 89.4 | 79.4 | 86.8 |
| **+ HAMLET (Ours)** | **52.5** | **65.4** | **66.4** | **99.0** | **100.0** | **99.2** | **92.2** | **97.6** |

Table 3: **Simulation benchmark results on CogACT.** We compare HAMLET with baseline methods on the SimplerEnv-Bridge benchmark. Each metric reports the success rate (%) on four WidowX tasks in SimplerEnv, with separate reporting for grasp success and full success. 'Avg.' denotes the average full success rate (%) across the four tasks, and all CogACT results are faithfully reproduced by us. **Bold** and underline indicate best and runner-up results, respectively.

| Method | Spoon on Towel | | Carrot on Plate | | Stack Block | | Eggplant in Basket | | |
| | Grasp | Success | Grasp | Success | Grasp | Success | Grasp | Success | Avg. |
|---|---|---|---|---|---|---|---|---|---|
| OpenVLA | 4.1 | 0.0 | 33.3 | 0.0 | 12.5 | 0.0 | 8.3 | 4.1 | 1.0 |
| Octo-Base | 34.7 | 12.5 | 52.8 | 8.3 | 31.9 | 0.0 | 66.7 | 43.1 | 16.0 |
| Octo-Small | 77.8 | 47.2 | 27.8 | 9.7 | 40.3 | 4.2 | 87.5 | 56.9 | 30.0 |
| RoboVLM | 54.2 | 29.2 | 25.0 | 25.0 | 45.8 | 12.5 | 58.3 | 58.3 | 31.3 |
| SpatialVLA | 20.8 | 16.7 | 29.2 | 25.0 | 62.5 | **29.2** | **100.0** | **100.0** | 42.7 |
| CogACT | 87.5 | 58.3 | 41.7 | 37.5 | 70.8 | 20.8 | 91.7 | 91.7 | 52.1 |
| + Multi-frame | 83.3 | 50.0 | 79.2 | 50.0 | 70.8 | 20.8 | 70.8 | 70.8 | 47.9 |
| **+ HAMLET (Ours)** | **91.7** | **75.0** | **83.3** | **62.5** | **75.0** | 16.7 | **100.0** | **100.0** | **63.5** |

default, we use moment tokens of length 4, a 2-layer Transformer as the memory module, and a history length of 4. Full hyperparameters and implementation details are provided in Section A.3.

## 4.2 MAIN RESULTS

**Real-world evaluation.** For real-world environment, we train each model for each task independently and evaluate their performance by averaging the success rates over 24 trials per task. For each task, partial success rates are also reported, where the criteria are: (i) *Pick-and-Place Once*: pick and place the cube at the correct site once; (ii) *Cover Cube*: cover the first cube with the nearest cup; and (iii) *Stage Cube*, stage the a cube to the auxiliary site. As shown in Table 1 and Figure 3, the base model (GR00T N1.5) struggles with these tasks, achieving only 12.5% success on Pick-and-Place Twice and often becoming confused about which direction to move (see more rollouts in Section B.2). Notably, applying HAMLET to GR00T N1.5 yields substantial improvements across all tasks, achieving an average improvement of 47.2% and underscoring its effectiveness in leveraging historical context.

**Simulation benchmarks.** To validate generalizability of HAMLET across generic benchmarks, we evaluate it on the standard simulation benchmarks, RoboCasa (Nasiriany et al., 2024) and LIBERO (Liu et al., 2023). As shown in Table 2, naïvely extending GR00T N1.5 with multi-frame inputs degrades the baseline performance by 3.3% in RoboCasa (100 demos) and 8.8% in LIBERO. This highlights an inherent weakness of this approach: (a) by conditioning only on consecutive frames during training, the model struggles to generalize to test environments with dynamically varying observations, and (b) merely accessing more past information can cause the policy to pick up spurious temporal correlations, a phenomenon also known as causal confusion (Wen et al., 2020; De Haan et al., 2019; Seo et al., 2023). On the other hands, HAMLET, when applied on top of GR00T N1.5, successfully improves performance across benchmarks: in LIBERO, it pushes the prior best score 95.6%, near-saturated success rate—up to 97.6%. This supports the advantage of our

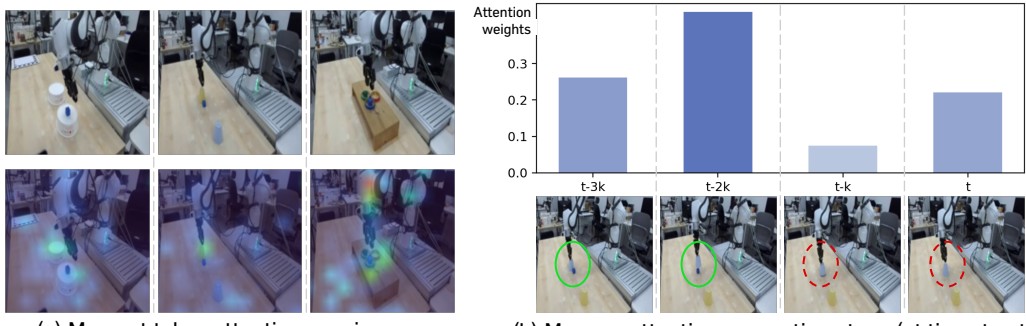

(a) Moment token attention over images     (b) Memory attention across timesteps (at timestep t)

Figure 4: **What does the memory network memorize?** (a) Visualization of self-attention map of moment tokens over input images inside the VLM, showing that they concentrate strongly on task-relevant regions. (b) Normalized self-attention weights of the memory module across the moment token sequence, indicating which timesteps contribute most to the memory features.

design: since HAMLET still receives single-frame inputs via external memory module, it effectively exploits historical context while preserving single-frame VLA's generalizability.

**Generalization to other VLAs.** We further validate the scalability of our framework in transforming existing VLAs into history-aware policy beyond GR00T N1.5. Specifically, we consider CogACT (Li et al., 2024a) as base model, which is another pre-trained VLA based on diffusion policy, on the Simpler-WidowX simulation benchmark (Walke et al., 2023). Table 3 reports both partial and full success rates across four tasks. Similar other simulation results (*e.g.*, Table 2), the multi-frame baseline fails to consistently improve success rate across tasks, possibly due to the its pool generalizability. In other hands, HAMLET still demonstrates clear improvements: it not only improve partial success rates across all tasks but, more importantly, significantly improves final task completion, achieving an average full success rate of 63.5%, compared to the original 52.1%. These highlight HAMLET's flexibility as an easily integrable fine-tuning framework, without the need for costly re-training.

### 4.3 MORE ANALYSIS

We further analyze the individual components in HAMLET and its efficiency over baselines. Throughout this section, unless otherwise specified, we consider the GR00T N1.5 on RoboCasa (100 demos).

**What does the memory module memorize?** We qualitatively analyze (a) how the proposed moment tokens encode information at each timestep and (b) how the memory module processes past information. First, we confirm that the moment token attends more to task-relevant parts that change over timesteps and less to static parts. Indeed, as shown in Figure 4(a), higher attention values of the moment token concentrate on the gripper and objects that are associated with task success, while lower attention values are assigned to background regions. This is possibly due to the initialization via time-contrastive loss (see Section 3.1), which encourages the tokens to extract distinguishable features over time. Next, we observe that the memory module selectively attends to past information depending on the context within the episode. As shown in Figure 4(b), in the *Cover-and-Stack* task, at the moment when it is necessary to decide which cup to approach after the cube is covered with a cup, the memory module assigns higher attention to the past timestep when the blue cube was previously visible. Additional qualitative results are provided in Section B.2.

**Efficiency analysis.** We analyze the efficiency of HAMLET by comparing with the multi-frame baseline which naïvely appends past observation frames, under varying history lengths. Specifically, we measure average latency (ms) and average peak GPU memory usage (MB) per environment timestep in the RoboCasa simulation. As shown in Table 4, the multi-frame baseline incurs substantial overhead in both metrics. For example, at a history length of 8, it requires roughly $2.4\times$ greater latency and $7\times$ higher memory than vanilla inference. In contrast, HAMLET shows only minimal overheads across history lengths, *e.g.*, $1.02\times$ at length 4 and $1.07\times$ at length 8. These results demonstrate the clear advantage of HAMLET over the multi-frame baseline in terms of efficiency.

Table 4: **Efficiency analysis.** Average latency and peak memory usage measured on RoboCasa datasets. Both metrics are computed at each timestep within an episode and then averaged. For fair comparison, memory for original VLA parameters is excluded, except for the memory module in HAMLET. All measurements were on an NVIDIA A100 GPU. ↓ indicates lower values are better.

| Method | History Length | Latency (ms, ↓) | Peak memory (MB, ↓) |
|---|---|---|---|
| GR00T N1.5 | 1 | 80.5 (1.00×) | 289 (1.00×) |
| + Multi-frame | 4 | 108.5 (1.35×) | 1051 (3.64×) |
| **+ HAMLET (Ours)** | 4 | **82.4 (1.02×)** | **566 (1.96×)** |
| + Multi-frame | 8 | 193.0 (2.40×) | 2023 (7.00×) |
| **+ HAMLET (Ours)** | 8 | **85.8 (1.07×)** | **578 (2.00×)** |

Table 5: **Ablation study.** Average success rate (%) on RoboCasa (100 demos) when selectively enabling different components of HAMLET. *Moment Concat.* concatenates all moment tokens without a memory module, whereas the Transformer-based memory yields the best overall performance.

| Moment Token | TCL | Memory Module | Avg. |
|---|---|---|---|
| ✗ | ✗ | ✗ | 62.6 |
| ✓ | ✗ | ✗ | 63.1 |
| ✓ | ✓ | ✗ | 63.4 |
| ✓ | ✗ | ✓ | 64.8 |
| ✓ | ✓ | ✓ | **65.4** |

(a) Component analysis.

| Token Length | Avg. |
|---|---|
| 1 | 64.3 |
| 4 | 65.4 |
| **8** | **66.4** |
| 16 | 65.9 |
| 32 | 62.7 |
| 64 | 62.5 |

(b) Moment token length.

| Method | Avg. |
|---|---|
| No Memory | 62.6 |
| Moment Concat. | 62.7 |
| RNN | 64.5 |
| LSTM | 65.0 |
| GRU | 64.3 |
| **Transformer** | **65.4** |

(c) Memory architecture.

**Ablation study.** We perform ablations on HAMLET to study the individual roles of its components, moment token length, and memory design. First, to assess the contribution of each component, we conduct an ablation study by removing key components one at a time: (i) the use of moment tokens, (ii) the initialization of Time-Contrastive Learning (TCL), and (iii) the presence of the memory module. In Table 5a, we confirm that removing the memory module leads to the largest performance drop, thereby confirming its critical role. We also observe that removing TCL-initialization consistently decreases performance, regardless of whether the memory module is used. Next, we examine the effect of moment token length (default = 4). Table 5b shows the performance improves as the length increases up to 8, but then gradually decreases beyond this point. Lastly, to understand the effect of design choices for memory module, we compare several architectures: (i) *Moment Concat.*, which naïvely concatenates past moment tokens without a memory module, (ii) RNN (Graves, 2012), LSTM (Hochreiter & Schmidhuber, 1997), and GRU (Cho et al., 2014), which are recurrent variants that utilize a single accumulated hidden state, (iii) Transformer, which we used in our experiments. From Table 5c, we find that the Transformer achieves the highest average success rate among the memory architectures. Interestingly, we observe that *Moment Concat.* yields almost no gains over the baseline, whereas other memory-based methods have shown performance gains.

**Generalization capability.** We investigate whether the memory module in our framework can transfer to unseen tasks, under the hypothesis that it learns generalizable knowledge for identifying which information from past timesteps is important, not limited to a specific dataset. To verify this, we first train the memory module and then freeze it while evaluating on different datasets. Specifically, we first train the memory module on LIBERO and then transfer it directly to RoboCasa. As shown in Table 6, even in this setup, HAMLET achieves a success

Table 6: **Generalization of memory module.** The memory module is trained with the dataset left of the arrow. A LIBERO-pretrained module provides gains for manipulation on RoboCasa, identifying that the learned memory representations can generalize across embodiment datasets.

| Method | Avg. |
|---|---|
| GR00T N1.5 | 62.6 |
| + HAMLET (LIBERO → RoboCasa) | 64.5 |
| + HAMLET (RoboCasa → RoboCasa) | **65.4** |

rate comparable to the in-distribution setting, where both training and evaluation are performed on RoboCasa. This demonstrates the practical flexibility of HAMLET to generalize across datasets, while eliminating the additional training cost for memory module adaptation.

## 5 CONCLUSION

In this work, we address the limitation of existing Vision-Language-Action models (VLAs), which typically rely on the current observation while ignoring past context. We propose HAMLET, a simple yet effective framework that enables pre-trained VLAs to leverage historical information without costly retraining from scratch. By introducing a lightweight memory module that integrates learnable moment tokens, HAMLET achieves significant improvements on long-horizon real-world tasks and standard simulation benchmarks, while maintaining computational efficiency. We hope our approach paves the way toward leveraging history-awareness in off-the-shelf VLA policies to tackle more complex robotic manipulation tasks reliably and effectively in practice.

## REPRODUCIBILITY STATEMENT

We provide full hyperparameter and implementation details in Section 4 and Section A.

## ACKNOWLEDGMENTS

This work was partly supported by Institute of Information & Communications Technology Planning & Evaluation (IITP) grant funded by the Korea government (MSIT) (No. RS-2019-II190075, Artificial Intelligence Graduate School Program (KAIST); No. RS-2025-02653113, High-Performance Research AI Computing Infrastructure Support at the 2 PFLOPS Scale; No. RS-2024-00509279, Global AI Frontier Lab). We are grateful to the RLWRLD Inc. for generously providing compute resources that supported a significant portion of the experiments conducted in this work. We also thank Jimin Lee and Heeseung Kwon for providing helpful support on conducting real-world experiments.

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

# A EXPERIMENTAL DETAILS

## A.1 MODEL DETAILS

In our experiments, we evaluate diffusion-based VLAs: GR00T N1 (Bjorck et al., 2025b)[1], GR00T N1.5 (Bjorck et al., 2025a)[2], $\pi_0$ (Black et al., 2025)[3], $\pi_0$-FAST (Pertsch et al., 2025)[4], and Co-gACT (Li et al., 2024a)[5]. All checkpoints are obtained from their official repositories.

## A.2 DATASETS

We evaluate HAMLET across both real and simulation environments, as demonstrated in Figure 5.

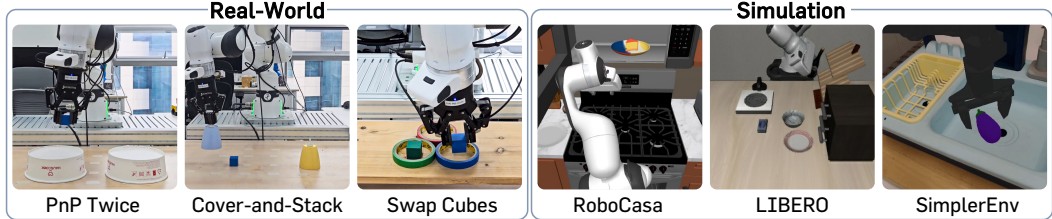

Figure 5: **Evaluation environments.** Left: Real-world evaluation comprises three tabletop tasks, which especially require awareness on historical context. Right: Simulation benchmarks include RoboCasa (Nasiriany et al., 2024) Kitchen, LIBERO (Liu et al., 2023), and SimplerEnv-Bridge (Li et al., 2024c), which consist of diverse indoor manipulation tasks.

**Real-world environment.** As shown in Figure 6, we use a Franka Research 3 robot arm equipped with a Robotiq 2F-85 gripper, following the DROID (Khazatsky et al., 2024) setup. Two camera views are provided: one mounted on the table and another on the wrist. On this real-robot platform, we design three handcrafted tabletop tasks as illustrated in Figure 7: (i) *Pick-and-Place Twice*, where the robot moves a cube between two sides twice; (ii) *Cover-and-Stack*, where the robot covers a cube with the nearest cup and then stacks another cup on top; and (iii) *Swap Cubes*, where the robot swaps the positions of two cubes using an auxiliary site. For each task, we collect 50 demonstrations for training, and report frame statistics in Table 7. Since trajectories are on average about 268 frames long, we evaluate each trial with a maximum limit of 700 timesteps: if the task has not been completed by then, the episode is counted as a failure.

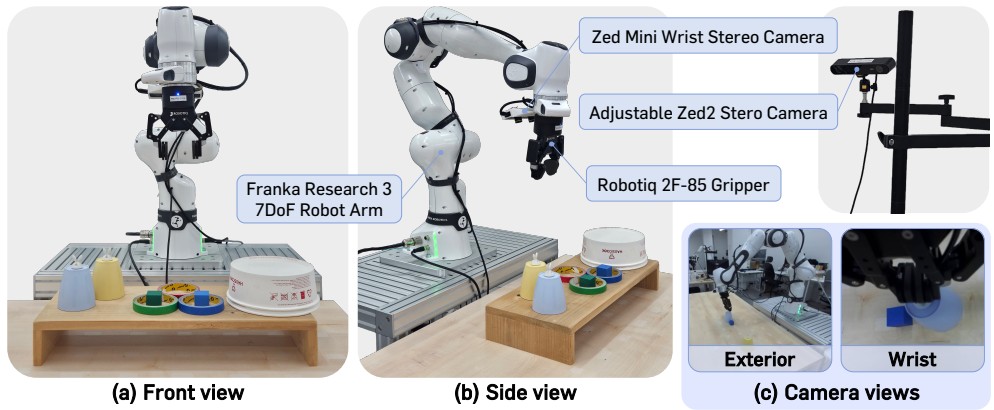

Figure 6: **Real-robot platform.** We specify the robot specifications and multi-camera views.

[1] https://huggingface.co/nvidia/GR00T-N1-2B
[2] https://huggingface.co/nvidia/GR00T-N1.5-3B
[3] gs://openpi-assets/checkpoints/pi0_base
[4] gs://openpi-assets/checkpoints/pi0_fast_base
[5] https://huggingface.co/CogACT/CogACT-Base

---

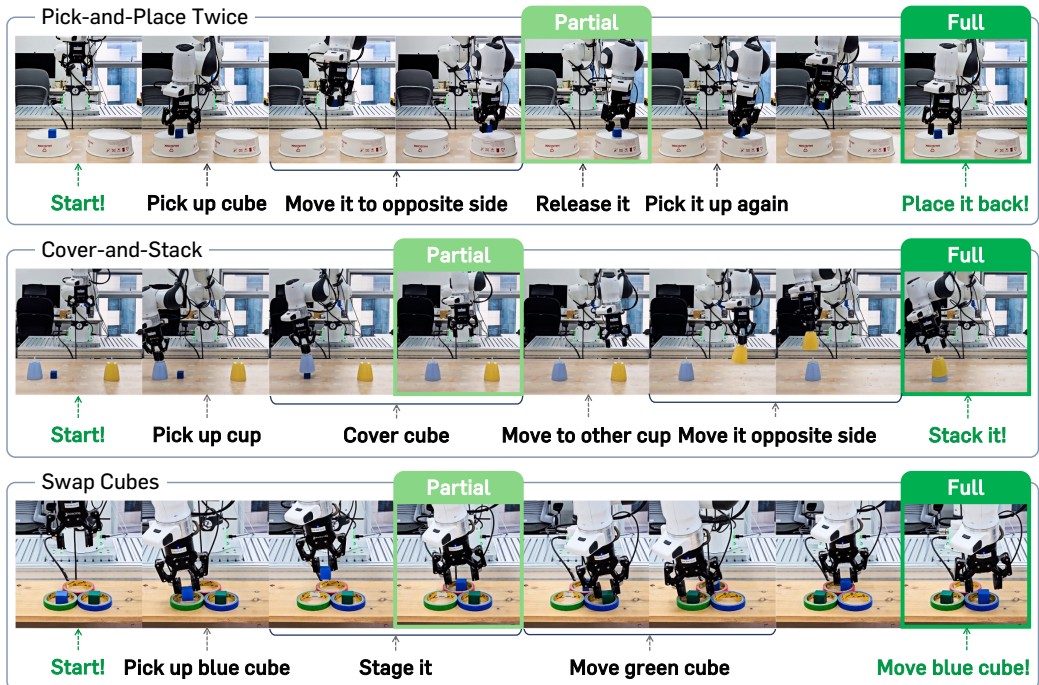

Figure 7: **Real-world tasks.** We design three history-dependent real-world tasks. Each partial criteria (*Pick-and-Place Once*, *Cover Cube*, *Stage Cube*) is described as the corresponding frame.

Table 7: **Dataset statistics for real-world tasks.** For each real-world dataset, we present the mean, maximum, and standard deviation of frame counts per episode. These statistics provide a basis for determining an appropriate maximum timestep limit during evaluation.

| Task | Mean #frames | Max #frames | Std #frames |
|------|-------------|-------------|-------------|
| Pick-and-Place Twice | 261 | 350 | 25.1 |
| Cover-and-Stack | 230 | 270 | 24.2 |
| Swap Cubes | 312 | 361 | 20.8 |

**Simulation benchmarks.** We evaluate HAMLET on three standard simulation benchmarks: (i) RoboCasa (Nasiriany et al., 2024), (ii) LIBERO (Liu et al., 2023) using GR00T N1.5, and (iii) SimplerEnv-Bridge (Walke et al., 2023) using CogACT. RoboCasa is a suite of kitchen manipulation tasks, comprising 24 tasks across 120 scenes. Following Bjorck et al. (2025a), we split the dataset into training and evaluation sets, evaluating each task over 50 episodes. LIBERO is designed to benchmark multi-task and lifelong robot learning, and includes 40 tasks grouped into four suites (Spatial, Object, Goal, and Long). Both RoboCasa and LIBERO use a Franka robot, and we adopt their officially released code for training and evaluation. Lastly, for SimplerEnv-Bridge, focused on real-to-simulation transfer, we use the BridgeV2 dataset (Walke et al., 2023) with a WidowX robot.

### A.3 IMPLEMENTATION

**Moment token and memory module.** Across experiments, we set the sequence length of moment tokens to 4 and use a 2-layer Transformer as the memory module. For the Transformer implementation, we adapt the LLaMA codebase from Hugging Face.[6] To initialize the moment tokens via Time-Contrastive Learning (TCL), we construct positive pairs using a combination of photometric distortions (brightness, contrast, and color jitter), Gaussian blur/noise, and small-patch occlusions applied stochastically to the same frame. Negative pairs are obtained by uniformly sampling frames

---

[6] https://huggingface.co/docs/transformers/en/model_doc/llama

that are more than 16 timesteps apart within the same trajectory, corresponding to the action horizon. We use a contrastive temperature of 0.07 for all TCL-based pre-training.

**Multi-frame baseline.** We follow the approach of multi-frame policies (Team et al., 2024; Liu et al., 2025) in handling historical observations to construct a naïve multi-frame baseline. For GR00T N1.5, we concatenate past observation frames across the three original views, *i.e.*, left, right, and wrist. For CogACT, since the original VLM is designed to process only a single view per timestep, we instead use a its cognition token and concatenate it to condition the action expert.

### A.4 TRAINING DETAILS

Based on the official training recipes (Bjorck et al., 2025a; Li et al., 2024a), we fine-tune GR00T N1.5 and CogACT.

- **GR00T N1.5**: Fine-tuned with the memory module for 60k steps using a batch size of 32 and a learning rate of 1e-5. Initialization of moment tokens with TCL is performed for up to 30k steps with a batch size of 64 and the same learning rate. During fine-tuning, both the VLM and moment tokens are kept frozen.
- **CogACT**: Fine-tuned with the memory module for 20k steps using a batch size of 32 and a learning rate of 2e-5. Initialization of moment tokens with TCL is performed for up to 30k steps with a batch size of 64 and the same learning rate, but training is terminated at 10k steps once the loss converges. During fine-tuning, both the VLM and moment tokens remain trainable.

In both cases, we use a default history length of 4. For all other baselines, including $\pi_0$, $\pi_0$-FAST, and GR00T N1, we also follow their official training codes.

### A.5 COMPUTATIONAL RESOURCES

- **GR00T N1.5**: Experiments is conducted on NVIDIA A100 80GB GPUs. Fine-tuning with the memory module on 4 GPUs requires ∼16 hours, comparable to the standard training time of ∼14 hours. Initialization of moment tokens with TCL adds ∼5 hours on 2 GPUs.
- **CogACT**: Experiments is conducted on NVIDIA H200 141GB GPUs. Fine-tuning with the memory module on 4 GPUs take ∼9 hours, compared to ∼4 hours for the standard setup. TCL-initialization requires ∼9 hours on 2 H100 GPUs.

We provide a summarization table detailing the modification costs (*i.e.*, model parameters, inference time, and training time) in Table 8.

Table 8: **Modification cost across different VLA types.** We report three metrics: model parameters, inference time, and training time. All inference was conducted on a single NVIDIA A100 GPU. Training was performed on 4 NVIDIA H200 GPUs, except for GR00T N1.5, whose VLM backbone was frozen during training and was therefore trained on 4 NVIDIA A100 GPUs.

| Method | Model parameters | Inference time (ms) | Training time (hours) |
|---|---|---|---|
| GR00T N1.5 | 2.72B | 80.5 (1.00×) | ∼ 14 |
| **+ HAMLET** | 2.86B | 82.4 (1.02×) | ∼ 16 |
| CogACT | 7.63B | 229.6 (1.00×) | ∼ 4 |
| **+ HAMLET** | 8.17B | 234.0 (1.01×) | ∼ 9 |

## B MORE EXPERIMENTAL RESULTS

### B.1 QUANTITATIVE RESULTS

We provide additional quantitative results to better understand the main components of HAMLET and its generalizability across various scenarios. We also present the complete RoboCasa metrics in Table 9, including the per-task success rates summarized in Table 2 for HAMLET and GR00T N1.5.

Table 9: **Full results on RoboCasa Kitchen with different dataset size.**

| Task (PnP = Pick-and-Place) | GR00T N1.5 | | | GR00T N1.5 + HAMLET | | |
|---|---|---|---|---|---|---|
| | 30 demos | 100 demos | 300 demos | 30 demos | 100 demos | 300 demos |
| Close Double Door | 46.0 | 80.0 | 90.0 | 58.0 | 82.0 | 82.0 |
| Close Drawer | 94.0 | 97.0 | 96.0 | 100.0 | 96.0 | 100.0 |
| Close Single Door | 88.0 | 94.0 | 98.0 | 96.0 | 94.0 | 96.0 |
| Coffee Press Button | 88.0 | 83.0 | 76.0 | 66.0 | 84.0 | 90.0 |
| Coffee Serve Mug | 62.0 | 63.0 | 58.0 | 70.0 | 68.0 | 76.0 |
| Coffee Setup Mug | 24.0 | 29.0 | 28.0 | 30.0 | 28.0 | 26.0 |
| Open Double Door | 70.0 | 76.0 | 84.0 | 68.0 | 86.0 | 96.0 |
| Open Drawer | 40.0 | 71.0 | 68.0 | 48.0 | 74.0 | 68.0 |
| Open Single Door | 54.0 | 69.0 | 78.0 | 58.0 | 82.0 | 84.0 |
| PnP from Cab to Counter | 22.0 | 51.0 | 50.0 | 26.0 | 46.0 | 54.0 |
| PnP from Counter to Cab | 36.0 | 49.0 | 50.0 | 46.0 | 48.0 | 56.0 |
| PnP from Counter to Microwave | 26.0 | 20.0 | 26.0 | 20.0 | 30.0 | 16.0 |
| PnP from Counter to Sink | 38.0 | 60.0 | 52.0 | 50.0 | 56.0 | 42.0 |
| PnP from Counter to Stove | 44.0 | 52.0 | 68.0 | 24.0 | 54.0 | 64.0 |
| PnP from Microwave to Counter | 22.0 | 40.0 | 44.0 | 24.0 | 24.0 | 24.0 |
| PnP from Sink to Counter | 60.0 | 60.0 | 74.0 | 54.0 | 56.0 | 56.0 |
| PnP from Stove to Counter | 38.0 | 70.0 | 66.0 | 56.0 | 74.0 | 76.0 |
| Turn Off Microwave | 62.0 | 96.0 | 96.0 | 88.0 | 98.0 | 100.0 |
| Turn Off Sink Faucet | 68.0 | 84.0 | 84.0 | 76.0 | 88.0 | 90.0 |
| Turn Off Stove | 12.0 | 18.0 | 30.0 | 12.0 | 20.0 | 28.0 |
| Turn On Microwave | 42.0 | 44.0 | 48.0 | 50.0 | 60.0 | 72.0 |
| Turn On Sink Faucet | 46.0 | 83.0 | 70.0 | 66.0 | 76.0 | 86.0 |
| Turn On Stove | 30.0 | 57.0 | 42.0 | 26.0 | 68.0 | 40.0 |
| Turn Sink Spout | 34.0 | 56.0 | 62.0 | 48.0 | 78.0 | 72.0 |
| Pick-and-Place | 35.8 | 50.3 | 53.8 | 37.5 | 48.5 | 48.5 |
| Open-or-Close | 65.3 | 81.2 | 85.7 | 71.3 | 85.7 | 87.7 |
| Others | 46.8 | 61.3 | 59.4 | 53.2 | 66.8 | 68.0 |
| **Average** | **47.8** | **62.6** | **64.1** | **52.5** | **65.4** | **66.4** |

Table 10: **Positive and negative pair construction for different TCL initialization methods.** We report the positive and negative pair constructions used by SimCLR, single-view TCN, multi-view TCN, and our TCL initialization.

| Method | Positive pair | Negative pair |
|---|---|---|
| SimCLR | Same view with classical image augmentations | In-batch samples |
| Single-view TCN | Same view at a nearby timestep | Same view at a far timestep |
| Multi-view TCN | Different view at the same timestep | Same view at a far timestep |
| TCL | Same view with classical image augmentations | Same view at a far timestep |

**Time-Contrastive Learning (TCL).** To understand the impact of individual parameters in TCL, we conduct an ablation study on different ways of constructing positive and negative pairs: (i) random initialization, (ii) SimCLR (Chen et al., 2020), (iii) single-view TCN (Sermanet et al., 2018), (iv) multi-view TCN (Sermanet et al., 2018), and (v) our TCL (default). Each method constructs positive and negative pairs from observation images, as summarized in Table 10. Table 11 shows that our TCL initialization achieves the highest average success rate among all methods, highlighting its effectiveness as a core component.

We further provide a more extensive analysis examining whether TCL influences final performance across datasets. As shown in Table 12, TCL consistently yields improvements across all datasets and even provides substantial gains in low-data scenarios (*i.e.*, up to +2.7% in RoboCasa 30 demos). These results highlight that TCL initialization plays a crucial role in enabling the moment tokens to effectively extract compact state features, especially when combined with other core components such as the memory module.

**More longer horizon scenarios.** To validate the effectiveness of HAMLET in more complex scenarios, we further evaluate it on a longer-horizon task, *Pick-and-Place Three Times*, which requires the model to return to the ready position after performing three consecutive pick-and-place operations.

Table 11: **Analysis on different TCL initialization methods.** We compare different TCL initialization by constructing positive and negative pairs. Our TCL method achieves the highest average success rate among all methods, demonstrating its effectiveness as a core component.

| Method | RoboCasa Kitchen | LIBERO | | | | |
| | 100 demo | Spatial | Object | Goal | Long | Avg. |
|---|---|---|---|---|---|---|
| Random initialization | 64.8 | **99.4** | 99.8 | 98.0 | 87.8 | 96.2 |
| SimCLR | 63.8 | **99.4** | 98.6 | 97.0 | 87.2 | 95.5 |
| Single-view TCN | 64.8 | 98.6 | 99.4 | 98.2 | 91.6 | 96.9 |
| Multi-view TCN | 64.9 | 99.0 | 99.0 | 97.8 | 90.2 | 96.5 |
| **TCL (Ours)** | **65.4** | 99.0 | **100.0** | **99.2** | **92.2** | **97.7** |

Table 12: **Effect of TCL.** We analyze the effect of TCL on RoboCasa Kitchen and LIBERO. For RoboCasa Kitchen, we report the average success rate (%) across 24 tasks with models trained using 30, 100, or 300 demonstrations per task. For LIBERO, each metric is the average success rate (%) across 10 tasks per suite, with training performed jointly on all suites. **Bold** indicates the best.

| Moment Token | TCL | Memory Module | RoboCasa Kitchen | | | LIBERO | | | | |
| | | | 30 | 100 | 300 | Spatial | Object | Goal | Long | Avg. |
|---|---|---|---|---|---|---|---|---|---|---|
| ✗ | ✗ | ✗ | 47.8 | 62.6 | 64.1 | 98.1 | 99.4 | 97.2 | 87.8 | 95.6 |
| ✓ | ✗ | ✓ | 49.8 | 64.8 | 65.1 | **99.4** | 99.8 | 98.0 | 87.8 | 96.2 |
| ✓ | ✓ | ✓ | **52.5** | **65.4** | **66.4** | 99.0 | **100.0** | **99.2** | **92.2** | **97.7** |

In particular, the policy needs to track how many successful pick-and-place operations have already been completed, requiring it to memorize more than 14 inference steps, whereas our method explicitly retains only 4 steps by default. As shown in Table 13, HAMLET achieves a significant improvement over the baseline on this long-horizon task, despite the required temporal dependency exceeding the original window size. This is possibly due to the causal nature of Transformers (Vaswani et al., 2017): the key–value pair of each token participates in the self-attention computation for all later tokens, influencing their hidden states and the corresponding key–value entries. Consequently, even after a past KV-cache entry is dropped, its information has already been integrated into the KV-cache of subsequent steps.

**Results on autoregressive VLAs.** We further validate the generalizability of HAMLET on autoregressive VLAs. Specifically, we apply it to OpenVLA (Kim et al., 2024) and $\pi_0$-FAST, two representative autoregressive VLAs built on action discretization. We evaluate OpenVLA on the LIBERO benchmark, while $\pi_0$-FAST is evaluated on the RoboCasa Kitchen benchmark (100 demos). As shown in Table 14, HAMLET consistently improves success rates over both baselines: for OpenVLA, achieving a 9.0% gain on LIBERO-Long, and for $\pi_0$-FAST, achieving a 5.8% gain on Pick-and-Place (PnP) tasks in RoboCasa. These results highlight that HAMLET effectively transforms pretrained VLAs into history-aware policies without relying on any specific architecture. For implementation, we fine-tune both models using a batch size of 32 for OpenVLA and 64 for $\pi_0$-FAST, following their official repositories[7][8] for other details.

## B.2 QUALITATIVE RESULTS

We provide additional qualitative results to better understand the main components of HAMLET.

**Time-Contrastive Learning (TCL).** To analyze the effect of Time-Contrastive Learning (TCL) on the moment token, we visualize the self-attention maps on RoboCasa, comparing randomly initialized tokens with TCL-initialized ones. As shown in Figure 8, randomly initialized moment tokens attend to sparsely distributed regions, including background areas that remain static across timesteps. In contrast, TCL-initialized moment tokens focus on more task-relevant regions related to successful

---

[7]https://huggingface.co/openvla/openvla-7b
[8]https://github.com/Physical-Intelligence/openpi

Table 13: **More longer horizon scenarios.** We validate HAMLET on the longer-horizon task *Pick-and-Place Three Times*, which requires tracking three consecutive pick-and-place cycles. We report the success rate (%, over 24 trials per task) and the average number of executed pick-and-place operations. **Bold** indicates the best.

| Method | Success rate | Executed PnPs |
|---|---|---|
| GR00T N1.5 | 8.3 | 1.042 |
| + HAMLET | **37.5** | **1.958** |

Table 14: **Results on autoregressive VLAs.** We evaluate HAMLET on two representative autoregressive models: (a) OpenVLA on the LIBERO benchmark, and (b) $\pi_0$-FAST on the RoboCasa Kitchen benchmark (100 demos). For LIBERO, we report the average success rate (%) across 10 tasks per suite, with training performed separately on each suite. For RoboCasa Kitchen, we report the average success rates (%) over 24 tasks, which are grouped into three categories: Pick-and-Place (PnP), Open-or-Close (OoC), and Others. **Bold** indicates the best.

(a) OpenVLA (Kim et al., 2024)

| | LIBERO | | | |
|---|---|---|---|---|
| Method | Spatial | Object | Goal | Long |
| OpenVLA | 84.8 | 62.6 | 71.6 | 48.6 |
| + HAMLET | **85.6** | **70.8** | **78.2** | **57.6** |

(b) $\pi_0$-FAST (Pertsch et al., 2025)

| | RoboCasa Kitchen | | |
|---|---|---|---|
| Method | PnP | OoC | Others |
| $\pi_0$-FAST | 17.0 | 60.7 | 46.6 |
| + HAMLET | **22.8** | **63.7** | **52.0** |

execution. For example, we observe that the object to be grasped by the gripper tends to receive higher attention values than other regions.

**Memory module.** As explored in Figure 4, we further provide examples of how our memory module attends to historical context. Specifically, we visualize the attention maps of the memory module during rollouts in the *Swap Cubes* task, where the robot swaps the blue cube with the green cube. In the left side of Figure 9, we confirm that the memory network attends to the key past timestep, which is crucial for determining the next action: for example, when deciding which cube to pick next, the memory module highly attends to the past timestep when the blue cube was previously placed down. Interestingly, the attention across past timesteps remains low when historical information is less critical, *e.g.*, during the initial move of the blue cube, as shown on the right side of Figure 9.

**Further example rollouts of real-world tasks.** We further illustrate rollouts by HAMLET and GR00T N1.5 for each real-world task in Figure 10, 11, 12. It can be clearly observed that GR00T N1.5 without history-awareness struggles with multi-step dependencies and often fails to recover from occlusions or ambiguous intermediate states, while HAMLET leverages its memory to better recognize the current state and complete tasks. In Figure 13, we further provide typical failure cases of naïve multi-frame baseline on GR00T N1.5, which often proceeds to the next actions while failing to recover from its failure state. This indicates that simply appending multi-frame inputs to VLAs might degrade their generalizability.

## C DISCUSSION

**Limitations.** While HAMLET maintains the efficiency of single-frame VLAs through its efficiency-oriented design, it still incurs additional training cost due to initialization with time-contrastive learning. Furthermore, although we demonstrate its applicability to recent diffusion-based VLAs, it may not directly extend to auto-regressive VLAs (Pertsch et al., 2025; Kim et al., 2024).

**Future directions.** One promising direction is to scale up HAMLET using larger-scale robotic manipulation datasets. As discussed in Table 6, each components of HAMLET, such as the memory module, can transfer effectively to unseen datasets. This suggests that initializing these components on large-scale datasets and then rapidly adapting them further enhance the practicality of our framework. In addition, while we adopted a relatively shallow Transformer for memory module to ensure scalability, a more exhaustive search over architectural parameters or the use of recent hybrid models (Lenz et al., 2025; Nano, 2025) is a straightforward approach for obtaining additional gains.

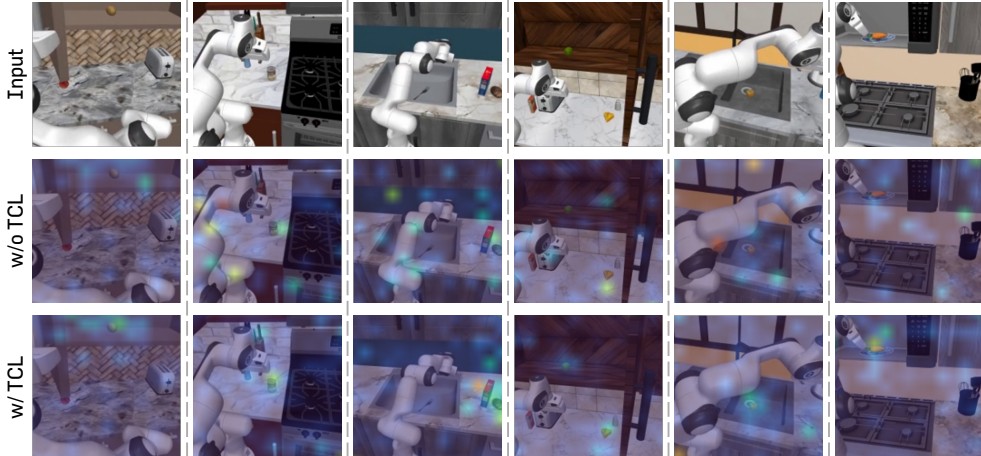

Figure 8: **Qualitative results of the moment tokens.** We analyze the self-attention of moment tokens on RoboCasa input frames, comparing random initialization (w/o TCL) with TCL-initialization (w/ TCL). After TCL-initialization, the moment tokens tend to concentrate more on task-relevant regions, such as the object to be grasped.

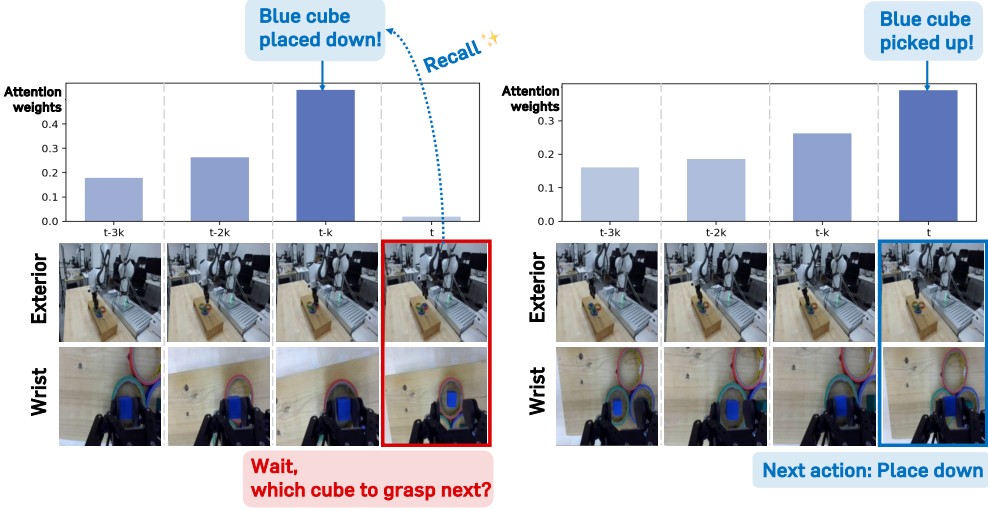

Figure 9: **Qualitative results of the memory module.** We analyze the self-attention of the memory module on the *Swap Cubes* task, where the robot swaps the blue cube with the green cube. We report two distinctive cases: (left) one where recalling a specific past timestep is necessary for the next action–such as when the blue cube has already been placed down before moving the green cube–and (right) another where recalling is less critical–such as during the initial move of the blue cube. We observe that our memory module clearly attends to the relevant past timestep when required.

## D    THE USE OF LARGE LANGUAGE MODELS

We acknowledge that large language models (LLMs) were used in the preparation of this manuscript to assist with writing quality. We mainly utilize them to find grammatical errors, suggest alternative vocabulary. All ideas, analyses, and conclusions presented in this paper are solely those of the authors.

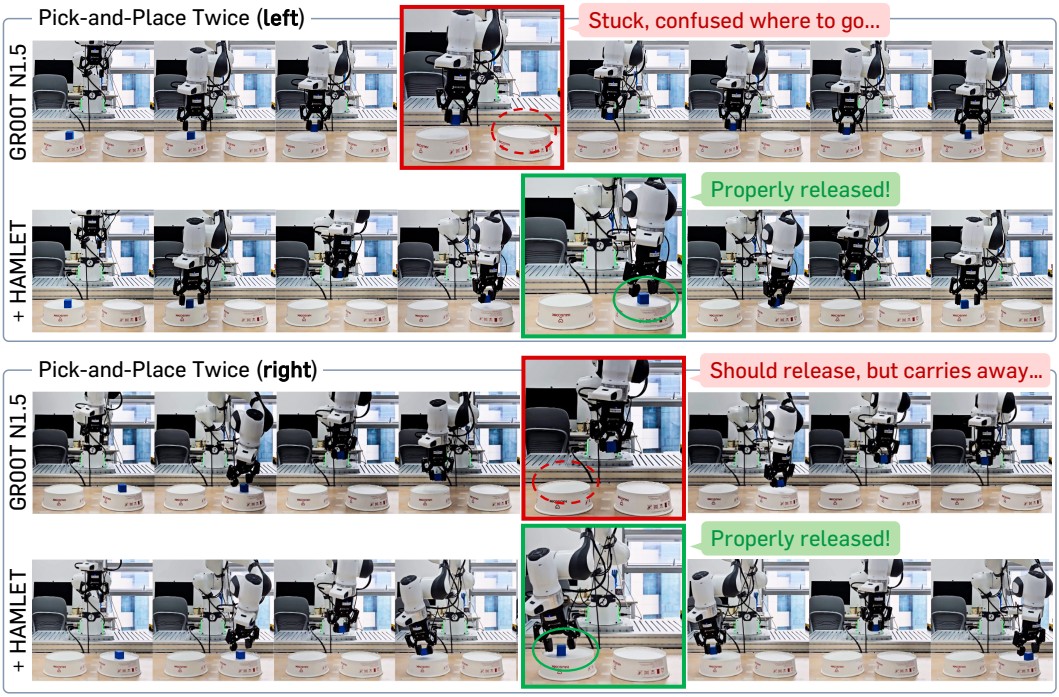

Figure 10: **Example rollouts of Pick-and-Place Twice task.**

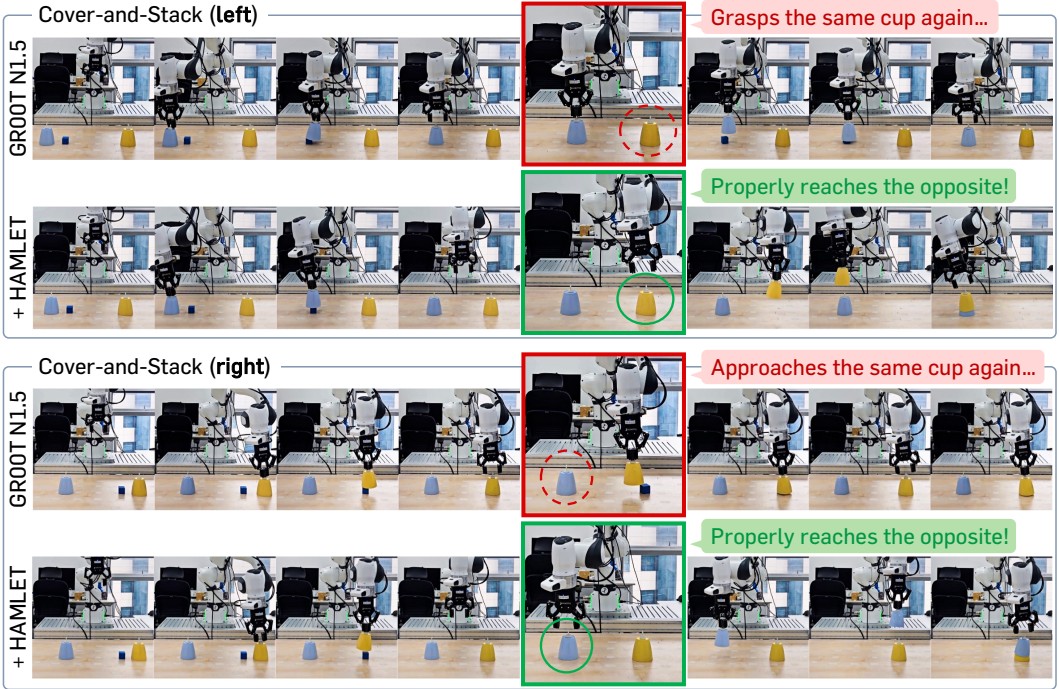

Figure 11: **Example rollouts of Cover-and-Stack task.**

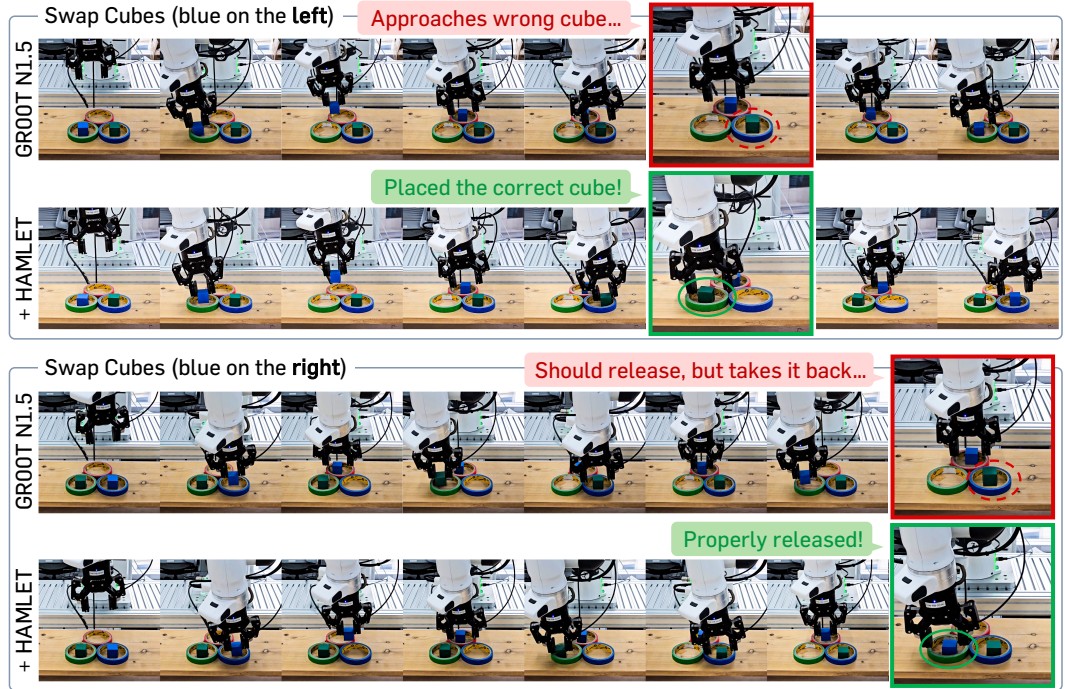

Figure 12: **Example rollouts of Swap Cubes task.**

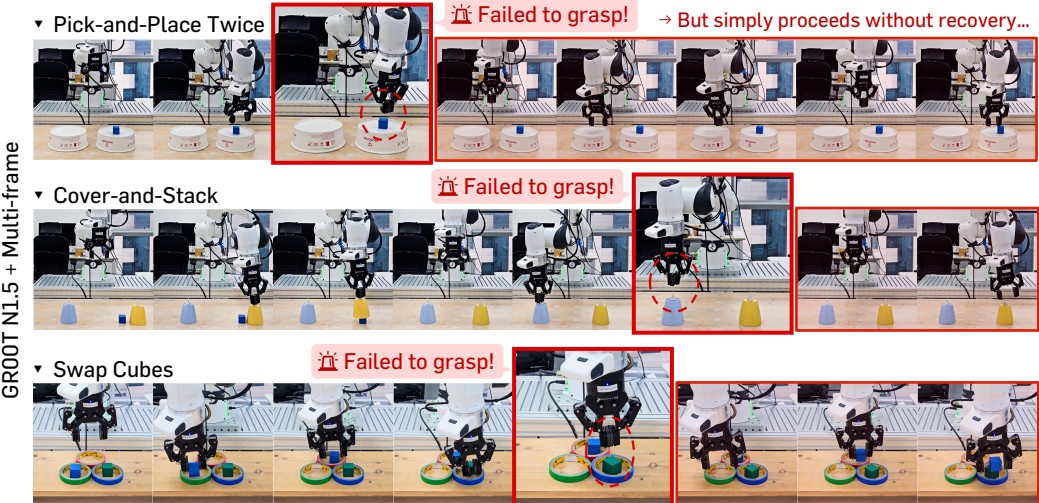

Figure 13: **Example rollouts by naïve multi-frame baseline.** We observe that the naïve multi-frame baseline often proceeds without recovery, merely copying action trajectories observed during training. This is possibly due to its poor generalizability, using only consecutive frames throughout training.

