# OpenReview forum: "HAMLET: Switch Your Vision-Language-Action Model into a History-Aware Policy"
_ICLR.cc/2026/Conference — ICLR 2026 Poster_

### Official Review · Reviewer_rLRn · 2025-10-28

**Soundness:** 3
**Presentation:** 3
**Contribution:** 3
**Rating:** 8
**Confidence:** 4

**Summary:**

The paper observes that multi-step robotic manipulation tasks are often history-dependent, whereas most existing Vision-Language-Action (VLA) models rely only on the current observation. To address this, the authors propose HAMLET, a framework that transforms pretrained VLAs into history-aware policies by introducing moment tokens, compact temporal representations learned through time-contrastive learning, and a lightweight memory module that aggregates past tokens to enhance action prediction.

**Strengths:**

1. The experimental results are very impressive.

2. This is a simple yet highly effective approach.

3. The writing is clear and well-structured.

**Weaknesses:**

See questions.

**Questions:**

1. Using past information is a very common strategy, as many previous imitation learning methods have incorporated historical observations [1,2]. Therefore, this aspect of the work feels somewhat less novel. However, the impressive results largely compensate for this. I am genuinely curious—why does adding the query token and aggregating past information lead to such a significant improvement? It would be great if the authors could provide more analysis on this.

2. As discussed in [1] and several other works, incorporating past information can lead to the copycat problem. Did you encounter this issue in your experiments?

3. In the simulation experiments, the multi-frame setting performs worse than the baseline. Why is that the case? Intuitively, having access to more information should improve performance, shouldn’t it?

[1] Fighting Copycat Agents in Behavioral Cloning from Multiple Observations. NeurIPS 2023.

[2] LIBERO: Benchmarking Knowledge Transfer for Lifelong Robot Learning. NeurIPS 2023.

---

> ### Author Response · Authors · 2025-11-19
>
> Dear reviewer rLRn,
>
> We sincerely appreciate your efforts and insightful comments to improve the manuscript.
>
> ---
>
> **[Q1-1] Novelty of HAMLET relative to prior methods that incorporate historical observations.**
>
> Although previous works do share a similarity in that they also incorporate historical observations into robot policies, their approaches remain limited because (a) they rely on simple policy architectures [1,2] (e.g., MLPs or RNNs) with limited generalizability, and (b) they must be trained from scratch [3,4], which imposes substantial computational costs. In contrast, our work addresses how to integrate historical context into large-scale pre-trained VLAs and provides a scalable solution that can be directly applied without re-training from scratch.
>
> [1] Wen et al., Fighting Copycat Agents in Behavioral Cloning from Observation Histories, NeurIPS 2020.
>
> [2] Seo et al., Regularized Behavior Cloning for Blocking the Leakage of Past Action Information, NeurIPS 2023.
>
> [3] Li et al., Towards Generalist Robot Policies: What Matters in Building Vision-Language-Action Models, preprint 2024.
>
> [4] Huang et al., OTTER: A Vision-Language-Action Model with Text-Aware Visual Feature Extraction, ICML 2025.
>
> **[Q1-2] Why HAMLET’s design leads to large performance improvements?**
>
> To further understand why HAMLET achieves significant performance improvements despite its simple design, we qualitatively analyze its inner behavior. We observe that (a) the moment tokens effectively compress VLM features at each timestep (e.g., Figure 8 in the Appendix), and (b) the memory module selectively attends to crucial past timesteps when necessary (e.g., Figure 9 in the Appendix). As shown in the rollout examples in Figures 10-12, these components enable HAMLET to more accurately predict the next action and better reliably identify failure states, resulting in more effective recovery behaviors compared to the baseline that naïvely leverages past frames (e.g., Figure 13 in the Appendix).
>
> ---
>
> **[Q2,3] Did you encounter copycat behavior in the experiments? And clarify why the multi-frame setting degrades baseline performance despite having access to additional past information.**
>
> Yes, we do observe the copycat problem [1] when past information is incorporated naïvely. In our experiments, simply providing additional historical frames leads to performance degradation or only marginal improvements (e.g., “Multi-frame” in Tables 2 and 3; “Moment Concat.” in Table 5(c)). As shown in Figure 13 in the Appendix, these baselines often execute a memorized next action instead of performing recovery. This suggests that merely having access to more information can cause the policy to pick up spurious temporal correlations, a phenomenon also known as causal confusion [1,5,6].
>
> In contrast, HAMLET mitigates this issue by compressing historical information into compact memory features through the moment tokens and the Transformer-based memory module. This information bottleneck prevents the policy from overfitting to low-level frame-to-frame correlations and enables it to use temporal context more robustly, avoiding the degradation typically caused in the multi-frame setting.
>
> [1] Wen et al., Fighting Copycat Agents in Behavioral Cloning from Observation Histories, NeurIPS 2020.
>
> [5] Haan et al., Causal Confusion in Imitation Learning, NeurIPS 2019.
>
> [6] Seo et al., Regularized Behavior Cloning for Blocking the Leakage of Past Action Information, NeurIPS 2023.
>
> ---
>
> We sincerely appreciate the time and effort you took to engage with our work. If there are any remaining concerns beyond those already discussed, we would be grateful if you could share them so that we may address them more thoroughly during the remainder of the rebuttal period.

---

> > ### Comment · Reviewer_rLRn · 2025-11-21
> >
> > The authors addressed all of my concerns. It is a simple yet effective method, and I will maintain my positive score.

---

> > > ### Author Response · Authors · 2025-11-26
> > >
> > > Dear Reviewer rLRn,
> > >
> > > We truly appreciate your acknowledgement of our rebuttal. As discussed, within the revised manuscript, we have expanded the related work section to include additional literature on history-integration methods and have further clarified the causes of performance degradation observed in the multi-frame baseline.
> > >
> > > If you have any further comments or suggestions, please let us know. We are committed to improving the quality of our work, and we value your feedback.
> > >
> > > Thank you very much,\
> > > Authors

---

### Official Review · Reviewer_JGoo · 2025-10-29

**Soundness:** 4
**Presentation:** 3
**Contribution:** 3
**Rating:** 8
**Confidence:** 4

**Summary:**

This paper introduces HAMLET, primarily addressing the memory challenges inherent in Vision-Language-Action (VLA) models. The authors propose the use of Memory Tokens combined with a time-contrastive learning objective applied to the output features. This approach aims to obtain a compressed representation of the VLA's current internal state. This compact feature can then be provided to an action expert, supplying essential historical information. The method is notably lightweight and efficient, achieving a significant reduction in memory overhead.

**Strengths:**

- The paper is easy to follow, and the figures are clear and highly informative.

- The motivation behind this work is compelling. Most current VLA models neglect the issue of memory, and their tested benchmarks are often simple, placing minimal demands on historical information. However, as the authors correctly illustrate, real-world tasks requiring memory are abundant. Addressing this class of problems is crucial for VLA deployment.

- The proposed method is simple yet effective, incurring a low training burden. Crucially, it can be easily applied to many existing pre-trained VLA models to significantly enhance their memory capabilities.

**Weaknesses:**

This paper is comprehensive and well-executed; I found no discernible weaknesses.

**Questions:**

The authors successfully demonstrate and test many real-world tasks that are heavily dependent on memory, where the performance improvement brought by HAMLET is expected. However, the method also yields performance gains on benchmarks like LIBERO and SIMPLER. Since these tasks are logically completable without reliance on historical information, why does HAMLET still provide a performance boost? Could the authors offer an analysis or hypothesis for this unexpected improvement on tasks that are ostensibly memory-agnostic?

---

> ### Author Response · Authors · 2025-11-19
>
> Dear reviewer JGoo,
>
> We sincerely appreciate your efforts and insightful comments to improve the manuscript.
>
> ---
>
> **[Q1] Why HAMLET improves performance on LIBERO and SIMPLER despite being memory-agnostic tasks?**
>
> We thank the reviewer for the insightful question. While tasks in LIBERO and SIMPLER can technically be completed from a single observation, relying solely on the current snapshot introduces inherent ambiguity about the underlying state. Without access to preceding context, the policy cannot fully determine whether it is progressing correctly, which leads it to repeat similar suboptimal behaviors, especially in failure cases. Meanwhile, historical information provides not only access to past states, but also richer contextual cues for correctly interpreting the current state. Empirically, we observe that HAMLET often identifies failure states more reliably and performs more effective recovery behaviors across all benchmarks.
>
> ---
>
> We sincerely appreciate the time and effort you took to engage with our work. If there are any remaining concerns beyond those already discussed, we would be grateful if you could share them so that we may address them more thoroughly during the remainder of the rebuttal period.

---

> > ### Comment · Reviewer_JGoo · 2025-11-25
> >
> > Thank you for your reply. Looking for more discussion.
> > Overall, I feel that the current explanation is not entirely convincing. I agree that historical context is indeed beneficial for the policy to understand whether it is progressing correctly and how it entered a suboptimal situation. However, as I notice, the LIBERO dataset is composed almost entirely of one-shot successful trajectories, lacking demonstrations of explicit error-correction or recovery behaviors. A history context where the robot "gets into trouble" would, in this case, be considered Out-of-Distribution (OOD) data, which I even suspect could have a negative impact on the policy.
> >
> > In general, I believe the understanding of history should ideally be acquired during the pre-training stage, where the data distribution is more diverse and encompasses a wider variety of states, including error-correction data. If the model is only fine-tuned on one-shot success demonstrations, I find it difficult to understand how the policy genuinely learns to correct errors based on history.
> >
> > This line of reasoning extends to the authors' claim of plug-and-play (or "easily integrable"). Given the nature of the fine-tuning data, do the authors view this as a current issue? Or does HAMLET possess a specific mechanism to alleviate this data distribution challenge?

---

> > > ### Author Response · Authors · 2025-11-26
> > >
> > > Dear Reviewer JGoo,
> > >
> > > Thank you for your insightful feedback. First, we agree with your point that including error-correction data can further benefit from history-aware training, especially in terms of recovery behaviors. However, we would like to point out that such gains are also feasible even at the fine-tuning scale. For instance, from our real-world experiments, we collect error-correction trajectories by continuing teleoperation until the end of an episode even when a failure occurs, where we observe that incorporating these trajectories leads to meaningful recovery behavior in the learned policy. Yet, we also believe adapting HAMLET with such error-correction data in the pre-training stage would be an interesting future direction.
> > >
> > > Regarding the concern that using past information at inference may introduce OOD issues when training on one-shot success datasets, we would like to clarify that HAMLET mitigates this limitation by leveraging past information as a complementary signal for robust action prediction. By transforming past observations into compact representations (i.e., moment token) and retrieving only informative moments (i.e., memory module), HAMLET leverages history as a useful auxiliary signal. Under inference-time deviations, this auxiliary signal helps the model to robustly rectify drift from the optimal trajectory. Consequently, the policy predicts more accurate action even under subtle failures (e.g., inaccurate gripper position before grasping a target object).
> > >
> > > Unlike HAMLET, we also note that naïvely incorporating all past information can suffer from OOD effects, often leading to performance degradation or only marginal improvements (e.g., "Multi-frame" in Tables 2 and 3; "Moment Concat." in Table 5(c)).
> > >
> > > If you have any further comments or suggestions, please let us know. We are committed to improving the quality of our work, and we value your feedback.
> > >
> > > Thank you very much,\
> > > Authors

---

### Official Review · Reviewer_oHKU · 2025-10-30

**Soundness:** 2
**Presentation:** 3
**Contribution:** 2
**Rating:** 4
**Confidence:** 3

**Summary:**

This article addresses the limitation of existing Vision-Language-Action models (VLAs) that rely solely on current observations while ignoring historical context. It proposes the HAMLET framework, which transforms VLAs into history-aware policies by compressing perceptual information at each timestep using **moment tokens** initialized via time-contrastive learning (TCL), and integrating these historical moment tokens to generate memory features through a lightweight Transformer memory module. The article conducts experiments in both real-world (three tabletop tasks) and simulation benchmarks (RoboCasa Kitchen, LIBERO, SimplerEnv-Bridge). Results show that HAMLET, built on GR00T N1.5, achieves an average success rate of 76.4% on history-dependent real-world tasks (surpassing the baseline by 47.2%), improves performance from 64.1% to 66.4% on RoboCasa Kitchen (100-demo setup), and from 95.6% to 97.7% on LIBERO. Additionally, it incurs only minimal computational overhead (1.02× latency and 1.96× peak memory usage at a history length of 4) and can be adapted to different VLA backbones such as GR00T N1.5 and CogACT, verifying the framework’s effectiveness and generalizability.

**Strengths:**

1. The article accurately identifies the core contradiction in robotic manipulation tasks—VLAs’ "single-frame dependency" versus the "history-dependent nature of tasks"—and addresses the pain point of high computational overhead in traditional multi-frame baselines. It directly responds to the demand for efficient history-aware policies in the field of robot learning, demonstrating clear research value.

2. The combined design of moment tokens and a lightweight memory module is ingenious. Moment tokens reduce redundant information storage through compression, while the Transformer-based memory module enables selective integration of historical information. Notably, the framework does not require large-scale retraining of pre-trained VLAs, meeting the practical "plug-and-play" requirement. The technical route is clear and innovative.

3. Experiments cover both real-world and simulation environments, including comparisons across multiple tasks, baselines, and VLA backbones. Further analyses—such as efficiency evaluations, ablation studies (on component contributions, token length, and memory architecture), and cross-dataset transfer experiments—validate the framework’s performance, efficiency, and generalizability. The detailed data and logical reasoning enhance the credibility of the conclusions1

**Weaknesses:**

1. Currently, HAMLET uses moment tokens (default length of 4) and a lightweight 2-layer Transformer memory module, which perform well in the article’s designed tabletop tasks and standard kitchen manipulation tasks. However, validation is lacking in more complex scenarios, such as multi-object interactions, dynamically disturbed environments, or long-horizon tasks (with a history length exceeding 8). The mentioned issue of "the lightweight historical feature compression network" is valid: while the lightweight design ensures efficiency, it may insufficiently capture historical information due to limited feature expression capabilities in complex scenarios. Additional experiments are needed to demonstrate the framework’s robustness in such contexts.

2. The article only verifies HAMLET’s performance on diffusion-based VLAs (GR00T N1.5, CogACT) and does not extend it to autoregressive VLAs (e.g., models proposed by Pertsch et al., 2025; Kim et al., 2024). It also fails to analyze the modification costs and performance differences when adapting the framework to different VLA types, limiting the breadth of arguments for the framework’s applicable scope.

3. the article mentions that TCL enhances the temporal discriminability of moment tokens but does not elaborate on how specific parameters of TCL—such as the intensity of augmentation methods and negative sample selection strategies—influence token quality. It also lacks visualizations or quantitative analyses comparing the feature expression capabilities of moment tokens under different initialization methods (e.g., random initialization, other self-supervised approaches), leading to insufficient justification for the design of this core component.

**Questions:**

see weakness

---

> ### Author Response · Authors · 2025-11-19
>
> Dear reviewer oHKU,
>
> We sincerely appreciate your efforts and insightful comments to improve the manuscript.
>
> ---
>
> **[W1] Concerns regarding the expressive capacity of the lightweight temporal module in more complex scenarios.**
>
> To address your concern, we evaluate our method on an additional longer-horizon task, “Pick-and-Place Three Times”, which requires the model to return to the ready position after performing three consecutive pick-and-place operations. Because the policy must track how many successful pick-and-place operations have already been executed, the required history length becomes much larger (over 14 inference steps, corresponding to 208 timesteps), while our default configuration explicitly uses only 4 inference steps (corresponding to 48 timesteps). As shown in the table below, HAMLET still achieves noticeable improvements over the baseline, suggesting that the compact moment tokens remain reasonably capable in more long, complex scenarios.
>
> \begin{array}{lcc}
> \hline
> \text{Method} & \text{Success rate (\\%)} & \text{Average number of executed PnPs} \newline
> \hline
> \text{GR00T N1.5} & \phantom{0}\text{8.3} & \text{1.042} \newline
> \text{+ HAMLET} & \text{\textbf{37.5}} & \text{\textbf{1.958}} \newline
> \hline
> \end{array}
>
> We also note that our qualitative analyses demonstrate that HAMLET effectively captures essential information in complex scenes across timestep despite its lightweight design. As shown in Figures 4, 8 and 9, the moment tokens consistently concentrate on task-relevant regions at each timestep, while the memory module selectively attends to the appropriate past frames when required. This suggests that focusing on essential cues allows the lightweight design to remain robust.
>
> ---
>
> **[W2-1] Evaluation limited to diffusion-based VLAs, not autoregressive VLAs.**
>
> We thank the reviewer for raising this important point regarding the breadth of VLA architectures evaluated in our work. In response, we additionally apply HAMLET to OpenVLA [1], a representative autoregressive VLA. We observe that HAMLET still improves success rates substantially over the baseline, achieving  6.2% average gain on LIBERO. This result highlights HAMLET’s generalizability, transforming existing VLAs into history-aware policies without dependence on specific VLA architectures.
>
> \begin{array}{l|ccccc}
> \hline
> & \rlap{~~~~~~~~~~~~~~~~~~~~~\text{LIBERO}} & & & & \newline
> \hline
> \text{Method} & \text{Spatial} & \text{Object} & \text{Goal} & \text{Long}
> & \textbf{Avg.} \newline
> \hline
> \text{OpenVLA[1]} & \text{84.8} & \text{62.6} & \text{71.6} & \text{48.6} & \text{66.9} \newline
> \text{+ HAMLET} & \textbf{85.6} & \textbf{70.8} & \textbf{78.2} & \textbf{57.6} & \textbf{73.1} \newline
> \hline
> \end{array}
>
> (We fine-tune both methods using a batch size of 32 and follow the official repositories for all other implementation details.)
>
> **[W2-2] Lack of analysis on modification costs across VLA types.**
>
> As requested, we also provide a summarization table detailing the modification costs (i.e., model parameters, inference time, and training time) across all VLA architectures used in our experiments. The comparison shows that HAMLET introduces only modest overhead while effectively attending to the historical context  across architectures.
>
> \begin{array}{l|ccc}
> \hline
> \text{Method} & \text{Model parameters} & \text{Inference time (ms)} & \text{Training time (hours)} \newline
> \hline
> \text{GR00T N1.5} & \text{2.72B} & \text{80.5 (1.00×)} & \text{$\sim$14} \newline
> \text{+ HAMLET} & \text{2.86B} & \text{82.4 (1.02×)} & \text{$\sim$16} \newline
> \hline
> \text{CogACT} & \text{7.63B} & \text{229.6 (1.00×)} & \text{$\sim$4} \newline
> \text{+ HAMLET} & \text{8.17B} & \text{234.0 (1.01×)} & \text{$\sim$9} \newline
> \hline
> \text{OpenVLA[1]} & \text{7.54B} & \text{242.5 (1.00×)} & \text{$\sim$10} \newline
> \text{+ HAMLET} & \text{8.08B} & \text{248.5 (1.02×)} & \text{$\sim$15} \newline
> \hline
> \end{array}
>
> (All inference was conducted on a single NVIDIA A100 GPU. Training was performed on 4 NVIDIA H200 GPUs, except for GR00T-N1.5, whose VLM backbone was frozen during training and which was therefore trained on 4 NVIDIA A100 GPUs.)
>
> [1] Kim et al., OpenVLA: An Open-Source Vision-Language-Action Model, CoRL 2024.

---

> ### Author Response · Authors · 2025-11-19
>
> **[W3] Insufficient analysis of TCL’s different design choices.**
>
> We agree that further clarification of TCL’s parameters would strengthen our statements. For constructing positive pairs, we apply (a) photometric distortions, (b) Gaussian blur/noise, and (c) occlusion with small black patches. For negative pairs, we uniformly sample frames that are more than 16 timesteps apart within the same trajectory, corresponding to the action horizon. We use a contrastive temperature of 0.07 and a batch size of 64 for all TCL-based pretraining.
>
> We also appreciate your suggestion to explore how different initialization methods influence the quality of moment tokens. In response, we conduct an extensive ablation study by varying the self-supervised (contrastive) initialization methods: (i) random-init, (ii) SimCLR [2], (iii) single-view TCN [3], (iv) multi-view TCN [3], and our (v) TCL. Each approach constructs positive and negative pairs using the observation images, as summarized in the following table:
>
> \begin{array}{lll}
> \hline
> \text{Method} & \text{Positive pair} & \text{Negative pair} \newline
> \hline
> \text{SimCLR [2]} & \text{Same view with classical image augmentations} & \text{In-batch samples} \newline
> \text{Single-view TCN [3]} & \text{Same view at a nearby timestep} & \text{Same view at a far timestep} \newline
> \text{Multi-view TCN [3]} & \text{Different view at the same timestep} & \text{Same view at a far timestep} \newline
> \text{TCL} & \text{Same view with classical image augmentations} & \text{Same view at a far timestep} \newline
> \hline
> \end{array}
>
> As shown in the table below, our TCL achieves the highest average success rate among all initialization methods, highlighting its effectiveness as a core component.
>
> \begin{array}{lcccccc}
> \hline
> & \text{RoboCasa Kitchen} & \rlap{~~~~~~~~~~~~~~~~~~~~~\text{LIBERO}} & & & & \newline
> \hline
> \text{Method} & \text{100demo} & \text{Spatial} & \text{Object} & \text{Goal} & \text{Long} & \text{Avg.} \newline
> \hline
> \text{Random-init.}
> & 64.8 & \textbf{99.4} & 99.8 & 98.0 & 87.8 & 96.2 \newline
> \text{SimCLR [2]}
> & 63.8 & \textbf{99.4} & 98.6 & 97.0 & 87.2 & 95.5 \newline
> \text{Single-view TCN [3]}
> & 64.8 & 98.6 & 99.4 & 98.2 & 91.6 & 96.9 \newline
> \text{Multi-view TCN [3]}
> & 64.9 & 99.0 & 99.0 & 97.8 & 90.2 & 96.5 \newline
> \text{TCL}
> & \textbf{65.4} & 99.0 & \textbf{100.0} & \textbf{99.2} & \textbf{92.2} & \textbf{97.7} \newline
> \hline
> \end{array}
>
> Additionally, Figure 8 in the Appendix provides a qualitative comparison between random initialization and TCL. We observe that, with TCL, the moment tokens consistently focus on task-relevant regions at each timestep, whereas randomly initialized tokens tend to attend to sparsely distributed and less meaningful regions.
>
> [2] Chen et al., A Simple Framework for Contrastive Learning of Visual Representations, ICML 2020.
>
> [3] Sermanet et al., Time-Contrastive Networks: Self-Supervised Learning from Video, ICRA 2018.
>
> ---
>
> We sincerely appreciate the time and effort you took to engage with our work. If there are any remaining concerns beyond those already discussed, we would be grateful if you could share them so that we may address them more thoroughly during the remainder of the rebuttal period.

---

> > ### Comment · Reviewer_oHKU · 2025-11-21
> > **Response to the rebuttal**
> >
> > Thanks for the detailed rebuttal.
> >
> > With these addtional experiments,  most of my concerns are solved.
> >
> > However, regarding the evaluation towards autoregressive VLAs,  I think it would be much better to add pi-0.5 except openvla.
> >
> > In this round, I would like to raise my score to 6, with strong willing to see the experiments on pi-0.5.

---

> ### Author Response · Authors · 2025-11-28
>
> Dear Reviewer oHKU,
>
> Thank you again for your constructive feedback. Regarding your suggestion to include Pi-0.5 [1] in addition to OpenVLA, we believe that you may have intended to refer instead to the autoregressive Pi0-FAST [2] model, as you cited in your initial comment.
>
> Following your suggestion, we have additionally applied HAMLET to Pi0-FAST. As shown in the table below, HAMLET still demonstrates clear improvements, achieving a 5.0% average gain on the RoboCasa 100-demo benchmark. We have also incorporated these results into the revised manuscript (see Table 14 in Appendix B.1).
>
> \begin{array}{lcccc}
> \hline
> & \rlap{~~~~~\text{RoboCasa Kitchen 100demo}} & & & & \newline
> \hline
> \text{Method} & \text{Pick-and-Place} & \text{Open-or-Close} & \text{Others} & \text{Avg.}\newline
> \hline
> \text{Pi0-FAST [2]}
> & 17.0 & 60.7 & 46.6 & 40.2 \newline
> \text{+ HAMLET}
> & \textbf{22.8} & \textbf{63.7} & \textbf{52.0} & \textbf{45.2} \newline
> \hline
> \end{array}
>
> We hope you find our response compelling enough. If you have any further comments or suggestions, please let us know. We are committed to improving the quality of our work, and we value your feedback.
>
> Thank you very much,\
> Authors
>
> [1] Physical Intelligence, π0.5 : a Vision-Language-Action Model with Open-World Generalization, preprint 2025.
>
> [2] Pertsch et al., FAST: Efficient Action Tokenization for Vision-Language-Action Models, preprint 2025.

---

> > ### Comment · Reviewer_oHKU · 2025-11-28
> > **second round response**
> >
> > OK. The additional experiments make sense. Thanks.

---

> > > ### Author Response · Authors · 2025-11-28
> > >
> > > Thank you sincerely for your constructive feedback and insightful suggestions. We truly appreciate your time and effort in reviewing our paper.

---

### Official Review · Reviewer_yS5A · 2025-11-02

**Soundness:** 3
**Presentation:** 3
**Contribution:** 2
**Rating:** 4
**Confidence:** 4

**Summary:**

This paper proposes HAMLET, a plug-and-play framework to add history-awareness to pre-trained VLAs without costly retraining. It introduces "moment tokens" to compress each timestep and a lightweight memory module to aggregate them. The tokens are notably initialized with Time-Contrastive Learning (TCL) to focus on dynamic cues. The method proves highly efficient and effective on medium-horizon tasks, outperforming naive multi-frame baselines.

**Strengths:**

- The framework is a highly efficient, plug-and-play module that demonstrates a significant advantage in computational cost (latency and memory) over naive multi-frame approaches.

- The method is validated extensively across multiple VLA backbones (GROOT N1.5, CogACT) and benchmarks (real-world, RoboCasa, LIBERO), proving its generalizability and effectiveness.

**Weaknesses:**

- The memory mechanism is a simple rolling-window Transformer, which is not scalable for truly long-horizon tasks as critical information will eventually be dropped from the fixed-size history, which does not consider the history before the rolling-window.
- The core methodological contribution of Time-Contrastive Learning (TCL) for token initialization provides only a marginal performance gain (0.6% in Table 5a) over a random initialization, questioning its overall necessity.
- The comparison against the 'Multi-frame' baseline is a weak strawman, as this naive frame-stacking approach is known to fail and interfere with current-state grounding; the paper lacks benchmarks against more sophisticated history integration techniques.
- The "plug-and-play" claim is misleading, as the framework requires a dedicated finetuning stage and a separate token initialization phase, which contradicts the common understanding of a plug-and-play module as a zero-shot, training-free component.

**Questions:**

See above.

---

> ### Author Response · Authors · 2025-11-19
>
> Dear reviewer yS5A,
>
> We sincerely appreciate your efforts and insightful comments to improve the manuscript.
>
> ---
>
> **[W1] Limitations of fixed-size history for longer-horizon tasks.**
>
> We would like to clarify that our memory module retains past information beyond the fixed history length, enabling generalization to truly longer-horizon tasks. This is possible due to the causal nature of Transformers [1]: the key-value pair of each token participates in the self-attention computation for all later tokens, influencing their hidden states and the corresponding key-value entries. Consequently, even after a past KV-cache entry is dropped, its information has already been integrated into the KV-cache of subsequent steps.
>
> To provide empirical evidence for this, we conduct an additional experiment on a longer-horizon task, “Pick-and-Place Three Times”. This task requires the model to return to the ready position after performing three consecutive pick-and-place operations, and the required memorization length (over 20s) far exceeds our fixed window of 48 timesteps (about 5s).
>
> As shown in the table below, our method achieves a noticeable improvement over the baseline on this extended task. This suggests that our history integration approach generalizes beyond the fixed window size, particularly when combined with KV-caching. We also note that our Transformer-based memory module can be readily combined with advanced KV-caching and length-extension techniques (e.g., RoPE [2], H2O [3]), which we believe have strong potential for further scaling to even longer history lengths.
>
> \begin{array}{lcc}
> \hline
> \text{Method} & \text{Success rate (\\%)} & \text{Average number of executed PnPs} \newline
> \hline
> \text{GR00T N1.5} & \phantom{0}\text{8.3} & \text{1.042} \newline
> \text{+ HAMLET (w/o KV-caching)} & \text{20.8} & \text{1.667} \newline
> \text{+ HAMLET (w/ KV-caching)} & \text{\textbf{37.5}} & \text{\textbf{1.958}} \newline
> \hline
> \end{array}
>
> [1] Vaswani et al., Attention is all you need, NIPS 2017.
>
> [2] Su et al., RoFormer: Enhanced Transformer with Rotary Position Embedding, preprint 2021.
>
> [3] Zhang et al., H2O: Heavy-Hitter Oracle for Efficient Generative Inference of Large Language Models, NeurIPS 2023.
>
> ---
>
> **[W2] Marginal performance gain from TCL.**
>
> We politely disagree with your concern regarding the necessity of TCL. As illustrated in Figure 8 in the Appendix, TCL initialization enables our moment token to concentrate more reliably on the important regions in the given observations, which results in effective compression of historical information for each timestep.
>
> Furthermore, to address your concern, we provide a more extensive analysis examining whether TCL influences final performance across datasets. As shown in the table below, TCL consistently yields improvements across all datasets and even provides substantial gains in low-data scenarios (i.e., up to +2.7% in RoboCasa-30demo). These results highlight that TCL initialization plays a crucial role in enabling the moment tokens to effectively extract compact state features, especially when combined with other core components such as the memory module.
>
> \begin{array}{ccc|ccc|ccccc}
> \hline
> \rlap{~~~~~~~~~~~~~~\text{Methods}} & & & \rlap{~~~~\text{RoboCasa Kitchen}} & & & \rlap{~~~~~~~~~~~~~~~~~~~~~\text{LIBERO}} & & & & \newline
> \hline
> \text{Moment Token} & \text{TCL} & \text{Memory Module}
> & \text{30demo} & \text{100demo} & \text{300demo}
> & \text{Spatial} & \text{Object} & \text{Goal} & \text{Long}
> & \textbf{Avg.} \newline
> \hline
> ✗ & ✗ & ✗
> & \text{47.8} & \text{62.6} & \text{64.1}
> & \text{98.1} & \text{99.4} & \text{97.2} & \text{87.8}
> & \text{95.6} \newline
> ✓ & ✗ & ✓
> & \text{49.8} & \text{64.8} & \text{65.1}
> & \textbf{99.4} & \text{99.8} & \text{98.0} & \text{87.8}
> & \text{96.2} \newline
> ✓ & ✓ & ✓
> & \textbf{52.5} & \textbf{65.4} & \textbf{66.4}
> & \text{99.0} & \textbf{100.0} & \textbf{99.2} & \textbf{92.2}
> & \textbf{97.7} \newline
> \hline
> \end{array}

---

> ### Author Response · Authors · 2025-11-19
>
> **[W3] “Multi-frame” baseline viewed as a weak strawman.**
>
> We would like to clarify that incorporating historical context into pre-trained VLAs is the first problem we address in this work. For example, OTTER [4] does leverage a multi-timestep context window, but it must be re-trained from scratch in that manner, which results in substantial computational cost for pretraining.
>
> Furthermore, to the best of our knowledge, there are no established techniques readily applicable to our setting (we would appreciate any specific suggestion if available). We consider the “Multi-frame” baseline to illustrate that simply leveraging historical context into existing VLAs is challenging in practice for two key reasons: (a) its substantial computational cost (Table 4), and (b) the potential for performance degradation (Tables 2 and 3) caused by redundant information in past frames, i.e., causal confusion [5,6,7]. We directly address both limitations by introducing an effective history integration technique based on the moment tokens and memory module.
>
> [4] Huang et al., OTTER: A Vision-Language-Action Model with Text-Aware Visual Feature Extraction, ICML 2025.
>
> [5] Haan et al., Causal Confusion in Imitation Learning, NeurIPS 2019.
>
> [6] Wen et al., Fighting Copycat Agents in Behavioral Cloning from Observation Histories, NeurIPS 2020.
>
> [7] Seo et al., Regularized Behavior Cloning for Blocking the Leakage of Past Action Information, NeurIPS 2023.
>
> ---
>
> **[W4] “Plug-and-play” claim viewed as misleading.**
>
> We have claimed our method as “plug-and-play” in the sense that it can be directly integrated into existing VLAs during the fine-tuning stage without requiring re-training from scratch. Nevertheless, we recognize that the term may lead to some confusion due to its broader implications, and we will revise the manuscript to clarify this point in the final draft.
>
> ---
>
> We sincerely appreciate the time and effort you took to engage with our work. If there are any remaining concerns beyond those already discussed, we would be grateful if you could share them so that we may address them more thoroughly during the remainder of the rebuttal period.

---

> > ### Comment · Reviewer_yS5A · 2025-11-25
> >
> > Thanks for the detailed response. I appreciate the clarification. However, I still have concerns about the use of "plug-and-play" terminology,  which typically refers to methods that can be directly applied without any training or fine-tuning (e.g., modular components that work out-of-the-box).  I suggest using more accurate terminology such as "easily integrated during fine-tuning" or "fine-tuning compatible" to avoid confusion.
> >
> > Overall, the method is simple and effective. The authors have adequately addressed my concerns in their rebuttal. I am raising my score to 6.

---

> > > ### Author Response · Authors · 2025-11-26
> > >
> > > Dear Reviewer yS5A,
> > >
> > > We truly appreciate your acknowledgement of our rebuttal. Following your suggestion, we have revised the manuscript to replace our "plug-and-play" terminology with more precise phrasing.
> > >
> > > If you have any further comments or suggestions, please let us know. We are committed to improving the quality of our work, and we value your feedback.
> > >
> > > Thank you very much,\
> > > Authors

---

### Author Response · Authors · 2025-11-25
**General Response**

Dear Reviewers and Area Chair,

We deeply appreciate your valuable time and effort spent reviewing our manuscript.

As the reviewers highlighted, our work addresses an important problem (Reviewers oHKU, JGoo) of pretrained VLAs: their reliance on only the current observation while ignoring historical context. Our framework, which consists of easily integrable tokens initialized with Time-Contrastive Learning (TCL) and a lightweight memory module, was recognized for both its novelty (Reviewers oHKU, JGoo) and effectiveness (All Reviewers), and it demonstrates strong generalizability (Reviewers yS5A, oHKU, JGoo) across both real-world and simulation environments.

In the rebuttal, we have carefully considered the reviewers' suggestions and addressed them in our responses. As a result, we have revised and enhanced the manuscript with the following additional discussions and experiments:

- Results on longer-horizon scenarios (Table 13 in Appendix B.1)
- Results on autoregressive VLAs (Table 14 in Appendix B.1)
- Ablation studies on TCL (Tables 10, 11, and 12 in Appendix B.1)
- Detailed parameter settings for TCL (Appendix A.3)
- A summarization table reporting modification cost across different VLA types (Table 8 in Appendix A.5)
- Related works on history-integration methods in robotics (Section 2 Related Works)
- Further clarification regarding the performance degradation of the multi-frame baseline (Section 4.2 Main Results)
- Updated terminology: replacing "plug-and-play" with "easily integrable"

In the revised manuscript, these updates are temporarily highlighted in $\textbf{\color{blue}blue}$ for your convenience.

We hope our responses and revisions sincerely address all the reviewers' concerns.

Thank you once again for your valuable contributions.

Warm regards,\
Authors

---

### Meta-Review · Area_Chair_a4kc · 2026-01-07

**Summary:**

The paper received strong endorsement from all four reviewers after rebuttal, who recognized its timely contribution to enabling history awareness in pre-trained vision-language-action models without full retraining. All these issues were directly and substantively addressed through new experiments, clearer explanations, and revised wording, leading two initially skeptical reviewers to raise their scores significantly.

**Reviewer Concerns:**

The authors comprehensively responded to every weakness and question raised by all reviewers.

**Reviewer Scores:**

Reviewer yS5A (initial 4) raised their score to 6 after the authors addressed all four weaknesses.
Reviewer oHKU (initial 4) raised their score to 6 following the addition of experiments.
Reviewer JGoo (initial 8) would maintain their score.
Reviewer rLRn (initial 8) would maintain their score.

---

### Decision · Program_Chairs · 2026-01-26

Accept (Poster)